# A Generic Formula and Some Special Cases for the Kullback–Leibler Divergence between Central Multivariate Cauchy Distributions

**DOI:** 10.3390/e24060838

**Published:** 2022-06-17

**Authors:** Nizar Bouhlel, David Rousseau

**Affiliations:** 1ImhorPhen Unit, UMR INRAe IRHS, Institut Agro Rennes-Angers, Université d’Angers, 42 Rue Georges Morel, 49070 Beaucouzé, France; 2LARIS, UMR INRAe IRHS, Université d’Angers, 62 Avenue Notre Dame du Lac, 49000 Angers, France; david.rousseau@univ-angers.fr

**Keywords:** Multivariate Cauchy distribution (MCD), Kullback–Leibler divergence (KLD), multiple power series, Lauricella D-hypergeometric series

## Abstract

This paper introduces a closed-form expression for the Kullback–Leibler divergence (KLD) between two central multivariate Cauchy distributions (MCDs) which have been recently used in different signal and image processing applications where non-Gaussian models are needed. In this overview, the MCDs are surveyed and some new results and properties are derived and discussed for the KLD. In addition, the KLD for MCDs is showed to be written as a function of Lauricella D-hypergeometric series $F_{D}^{\left(p\right)}$. Finally, a comparison is made between the Monte Carlo sampling method to approximate the KLD and the numerical value of the closed-form expression of the latter. The approximation of the KLD by Monte Carlo sampling method are shown to converge to its theoretical value when the number of samples goes to the infinity.

## 1. Introduction

Multivariate Cauchy distribution (MCD) belongs to the elliptical symmetric distributions [1] and is a special case of the multivariate *t*-distribution [2] and the multivariate stable distribution [3]. MCD has been recently used in several signal and image processing applications for which non-Gaussian models are needed. To name a few of them, in speckle denoizing, color image denoizing, watermarking, speech enhancement, among others. Sahu et al. in [4] presented a denoizing method for speckle noise removal applied to a retinal optical coherence tomography (OCT) image. The method was based on the wavelet transform where the sub-bands coefficients were modeled using a Cauchy distribution. In [5], a dual tree complex wavelet transform (DTCWT)-based despeckling algorithm was proposed for synthetic aperture radar (SAR) images, where the DTCWT coefficients in each subband were modeled with a multivariate Cauchy distribution. In [6], a new color image denoizing method in the contourlet domain was suggested for reducing noise in images corrupted by Gaussian noise where the contourlet subband coefficients were described by the heavy-tailed MCD. Sadreazami et al. in [7] put forward a novel multiplicative watermarking scheme in the contourlet domain where the watermark detector was based on the bivariate Cauchy distribution and designed to capture the across scale dependencies of the contourlet coefficients. Fontaine et al. in [8] proposed a semi-supervised multichannel speech enhancement system where both speech and noise follow the heavy-tailed multi-variate complex Cauchy distribution.

Kullback–Leibler divergence (KLD), also called relative entropy, is one of the most fundamental and important measures in information theory and statistics [9,10]. KLD was first introduced and studied by Kullback and Leibler [11] and Kullback [12] to measure the divergence between two probability mass functions in the case of discrete random variables and between two univariate or multivariate probability density functions in the case of continuous random variables. In the literature, numerous entropy and divergence measures have been suggested for measuring the similarity between probability distributions, such as Rényi [13] divergence, Sharma and Mittal [14] divergence, Bhattacharyya [15,16] divergence and Hellinger divergence measures [17]. Other general divergence families have been also introduced and studied like the ϕ-divergence family of divergence measures defined simultaneously by Csiszár [18] and Ali and Silvey [19] where the KLD measure is a special case, the Bregman family divergence [20], the R-divergences introduced by Burbea and Rao [21,22,23], the statistical *f*-divergences [24,25] and recently the new family of a generalized divergence called the (h,ϕ)-divergence measures introduced and studied in Menéndez et al. [26]. Readers are referred to [10] for details about these divergence family measures.

KLD has a specific interpretation in coding theory [27] and is therefore the most popular and widely used as well. Since information theoretic divergence and KLD in particular are ubiquitous in information sciences [28,29], it is therefore important to establish closed-form expressions of such divergence [30]. An analytical expression of the KLD between two univariate Cauchy distributions was presented in [31,32]. To date, the KLD of MCDs has no known explicit form, and it is in practice either estimated using expensive Monte Carlo stochastic integration or approximated. Monte Carlo sampling can efficiently estimate the KLD provided that a large number of independent and identically distributed samples is provided. Nevertheless, Monte Carlo integration is a too slow process to be useful in many applications. The main contribution of this paper is to derive a closed-form expression for the KLD between two central MCDs in a general case to benchmark future approaches while avoiding approximation using expensive Monte Carlo (MC) estimation techniques. The paper is organized as follows. Section 2 introduces the MCD and the KLD. Section 3 gives some definitions and propositions related to a multiple power series used to compute the closed-form expression of the KLD between two central MCDs. In Section 4 and Section 5, expressions of some expectations related to the KLD are developed by exploiting the propositions presented in the previous section. Section 6 demonstrates some final results on the KLD computed for the central MCD. Section 7 presents some particular results such as the KLD for the univariate and the bivariate Cauchy distribution. Section 8 presents the implementation procedure of the KLD and a comparison with Monte Carlo sampling method. A summary and some conclusions are provided in the final section.

## 2. Multivariate Cauchy Distribution and Kullback–Leibler Divergence

Let X be a random vector of $R^{p}$ which follows the MCD, characterized by the following probability density function (pdf) given as follows [2]
(1)$$\begin{array}{cc} \; & \;f_{X}\left(x|\mu ,\Sigma ,p\right)=\frac{\Gamma (\frac{1+p}{2})}{\pi^{\frac{p}{2}}\Gamma \left(\frac{1}{2}\right)}\frac{1}{{\left|\Sigma \right|}^{\frac{1}{2}}}\frac{1}{{\left[1+{\left(x-\mu \right)}^{T}\Sigma^{-1}\left(x-\mu \right)\right]}^{\frac{1+p}{2}}}. \end{array}$$
This is for any $x\in R^{p}$, where *p* is the dimensionality of the sample space, μ is the location vector, Σ is a symmetric, positive definite (p×p) scale matrix and Γ(.) is the Gamma function. Let $X^{1}$ and $X^{2}$ be two random vectors that follow central MCDs with pdfs $f_{X^{1}}\left(x|\Sigma_{1},p\right)=f_{X^{1}}\left(x|0,\Sigma_{1},p\right)$ and $f_{X^{2}}\left(x|\Sigma_{2},p\right)=f_{X^{2}}\left(x|0,\Sigma_{2},p\right)$ given by (Equation 1). KLD provides an asymmetric measure of the similarity of the two pdfs. Indeed, the KLD between the two central MCDs is given by
(2)$$\begin{array}{cc} \mathrm{KL}(X^{1}||X^{2}) & =\int_{R^{p}}\ln\left(\frac{f_{X^{1}}\left(x|\Sigma_{1},p\right)}{f_{X^{2}}\left(x|\Sigma_{2},p\right)}\right)f_{X^{1}}\left(x|\Sigma_{1},p\right)dx \end{array}$$
(3)$$\begin{array}{cc} \; & =E_{X^{1}}\left\{\ln f_{X^{1}}\left(X\right)\right\}-E_{X^{1}}\left\{\ln f_{X^{2}}\left(X\right)\right\}. \end{array}$$
Since the KLD is the relative entropy defined as the difference between the cross-entropy and the entropy, we have the following relation:(4)$$\begin{array}{cc} \mathrm{KL}(X^{1}||X^{2}) & =H\left(f_{X^{1}},f_{X^{2}}\right)-H\left(f_{X^{1}}\right) \end{array}$$
where $H\left(f_{X^{1}},f_{X^{2}}\right)=-E_{X^{1}}\left\{\ln f_{X^{2}}\left(X\right)\right\}$ denotes the cross-entropy and $H\left(f_{X^{1}}\right)=-E_{X^{1}}\left\{\ln f_{X^{1}}\left(X\right)\right\}$ the entropy. Therefore, the determination of KLD requires the expression of the entropy and the cross-entropy. It should be noted that the smaller $\mathrm{KL}(X^{1}||X^{2})$, the more similar are $f_{X^{1}}\left(x|\Sigma_{1},p\right)$ and $f_{X^{2}}\left(x|\Sigma_{2},p\right)$. The symmetric KL similarity measure between $X^{1}$ and $X^{2}$ is $d_{\mathrm{KL}}\left(X^{1},X^{2}\right)=\mathrm{KL}(X^{1}\left|\right|X^{2})+\mathrm{KL}(X^{2}\left|\right|X^{1})$. In order to compute the KLD, we have to derive the analytical expressions of $E_{X^{1}}\left\{\ln f_{X^{1}}\left(X\right)\right\}$ and $E_{X^{1}}\left\{\ln f_{X^{2}}\left(X\right)\right\}$ which depend, respectively, on $E_{X^{1}}\left\{\ln\left[1+X^{T}\Sigma_{1}^{-1}X\right]\right\}$ and $E_{X^{1}}\left\{\ln\left[1+X^{T}\Sigma_{2}^{-1}X\right]\right\}$. Consequently, the closed-form expression of the KLD between two zero-mean MCDs is given by
(5)$$\begin{array}{cc} \mathrm{KL}(X^{1}||X^{2}) & =\frac{1}{2}\log\frac{|\Sigma_{2}|}{|\Sigma_{1}|}-\frac{1+p}{2}\left(E_{X^{1}}\left\{\ln\left[1+X^{T}\Sigma_{1}^{-1}X\right]\right\}-E_{X^{1}}\left\{\ln\left[1+X^{T}\Sigma_{2}^{-1}X\right]\right\}\right). \end{array}$$
To provide the expression of these two expectations, some tools based on the multiple power series are required. The next section presents some definitions and propositions used for this goal.

## 3. Definitions and Propositions

This section presents some definitions and exposes some propositions related to the multiple power series used to derive the closed-form expression of the expectation $E_{X^{1}}\left\{\ln\left[1+X^{T}\Sigma_{1}^{-1}X\right]\right\}$ and $E_{X^{1}}\left\{\ln\left[1+X^{T}\Sigma_{2}^{-1}X\right]\right\}$, and as a consequence the KLD between two central MCDs.

**Definition** **1.**
*The Humbert series of n variables, denoted $\Phi_{2}^{\left(n\right)}$, is defined for all $x_{i}\in C,\;i=1,\ldots ,n$, by the following multiple power series (Section 1.4 in [33])*

(6)
$$\begin{array}{cc} \; & \Phi_{2}^{\left(n\right)}\left(b_{1},\ldots ,b_{n};c;x_{1},\ldots ,x_{n}\right)=\sum_{m_{1}=0}^{\infty }..\sum_{m_{n}=0}^{\infty }\frac{{\left(b_{1}\right)}_{m_{1}}\ldots {\left(b_{n}\right)}_{m_{n}}}{{\left(c\right)}_{\sum_{i=1}^{n}m_{i}}}\prod_{i=1}^{n}\frac{x_{i}^{m_{i}}}{m_{i}!}. \end{array}$$



The Pochhammer symbol ${\left(q\right)}_{i}$ indicates the *i*-th rising factorial of *q*, i.e., for an integer i>0
(7)$$\begin{array}{c} {\left(q\right)}_{i}=q\left(q+1\right)\ldots \left(q+i-1\right)=\prod_{k=0}^{i-1}\left(q+k\right)=\frac{\Gamma (q+i)}{\Gamma (q)} \end{array}$$

### 3.1. Integral Representation for $\Phi_{2}^{\left(n\right)}$

**Proposition** **1.**
*The following integral representation is true for $\mathrm{Real}\left\{c\right\}>\mathrm{Real}\left\{\sum_{i=1}^{n}b_{i}\right\}>0$ and $\mathrm{Real}\{b_{i}\}>0$ where Real{.} denotes the real part of the complex coefficients*

(8)
$$\begin{array}{cc} \; & \int \;\ldots \;\int_{\Delta }{\left(1-\sum_{i=1}^{n}u_{i}\right)}^{c-\sum_{i=1}^{n}b_{i}-1}\prod_{i=1}^{n}u_{i}^{b_{i}-1}e^{x_{i}u_{i}}du_{i}=B\left(b_{1},\ldots ,b_{n},c-\sum_{i=1}^{n}b_{i}\right)\Phi_{2}^{\left(n\right)}\left(b_{1},\ldots ,b_{n};c;x_{1},\ldots ,x_{n}\right) \end{array}$$

*where $\Delta =\{\left(u_{1},\ldots ,u_{n}\right)|0\leq u_{i}\leq 1,i=1,\ldots ,n;\;0\leq u_{1}+\ldots +u_{n}\leq 1\}$ and the multivariate beta function B is the extension of beta function to more than two arguments (called also Dirichlet function) defined as (Section 1.6.1 in [34])*

(9)
$$\begin{array}{c} B\left(b_{1},\ldots ,b_{n},b_{n+1}\right)=\frac{\prod_{i=1}^{n+1}\Gamma \left(b_{i}\right)}{\Gamma (\sum_{i=1}^{n+1}b_{i})}. \end{array}$$



**Proof.** The power series of exponential function is given by
(10)$$\begin{array}{c} e^{x_{i}u_{i}}=\sum_{m_{i}=0}^{\infty }\frac{x_{i}^{m_{i}}}{m_{i}!}u_{i}^{m_{i}}. \end{array}$$
By substituting the expression of the exponential into the multiple integrals we have
(11)$$\begin{array}{cc} \; & \int \;..\;\int_{\Delta }{\left(1-\sum_{i=1}^{n}u_{i}\right)}^{c-\sum_{i=1}^{n}b_{i}-1}\prod_{i=1}^{n}u_{i}^{b_{i}-1}e^{x_{i}u_{i}}du_{i}\\ \; & =\int \;..\;\int_{\Delta }{\left(1-\sum_{i=1}^{n}u_{i}\right)}^{c-\sum_{i=1}^{n}b_{i}-1}\left(\prod_{i=1}^{n}\sum_{m_{i}=0}^{\infty }\frac{x_{i}^{m_{i}}}{m_{i}!}u_{i}^{m_{i}+b_{i}-1}du_{i}\right)\\ \; & =\sum_{m_{1}=0}^{\infty }..\sum_{m_{n}=0}^{\infty }\left(\prod_{i=1}^{n}\frac{x_{i}^{m_{i}}}{m_{i}!}\right)\times I_{D} \end{array}$$
where the multivariate integral $I_{D}$, which is a generalization of a beta integral, is the type-1 Dirichlet integral (Section 1.6.1 in [34]) given by
(12)$$\begin{array}{cc} I_{D} & =\int \;\ldots \;\int_{\Delta }{\left(1-\sum_{i=1}^{n}u_{i}\right)}^{c-\sum_{i=1}^{n}b_{i}-1}\prod_{i=1}^{n}u_{i}^{m_{i}+b_{i}-1}du_{i}\\ \; & =\frac{\prod_{i=1}^{n}\Gamma \left(b_{i}+m_{i}\right)\Gamma \left(c-\sum_{i=1}^{n}b_{i}\right)}{\Gamma (c+\sum_{i=1}^{n}m_{i})}. \end{array}$$
Knowing that $\Gamma \left(b_{i}+m_{i}\right)=\Gamma \left(b_{i}\right){\left(b_{i}\right)}_{m_{i}}$, the expression of $I_{D}$ can be written otherwise
(13)$$\begin{array}{cc} \; & I_{D}=\frac{\prod_{i=1}^{n}\Gamma \left(b_{i}\right)\Gamma \left(c-\sum_{i=1}^{n}b_{i}\right)}{\Gamma (c)}\frac{\prod_{i=1}^{n}{\left(b_{i}\right)}_{m_{i}}}{{\left(c\right)}_{\sum_{i=1}^{n}m_{i}}}. \end{array}$$
Finally, plugging (Equation 13) back into (Equation 11) leads to the final result
(14)$$\begin{array}{cc} \; & \frac{\Gamma \left(c-\sum_{i=1}^{n}b_{i}\right)\prod_{i=1}^{n}\Gamma \left(b_{i}\right)}{\Gamma (c)}\sum_{\begin{array}{c} m_{1},\ldots ,\\ m_{n}=0 \end{array}}^{+\infty }\frac{\prod_{i=1}^{n}{\left(b_{i}\right)}_{m_{i}}}{{\left(c\right)}_{\sum_{i=1}^{n}m_{i}}}\prod_{i=1}^{n}\frac{x_{i}^{m_{i}}}{m_{i}!}=B\left(b_{1},\ldots ,b_{n},c-\sum_{i=1}^{n}b_{i}\right)\Phi_{2}^{\left(n\right)}\left(b_{1},\ldots ,b_{n};c;x_{1},\ldots ,x_{n}\right) \end{array}$$
□

Given Proposition 1, we consider the particular cases n={1,2} one by one as follows:

Case n=1
(15)$$\begin{array}{cc} \; & \frac{1}{B(b_{1},c-b_{1})}\int_{0}^{1}u_{1}^{b_{1}-1}e^{x_{1}u_{1}}{\left(1-u_{1}\right)}^{c-b_{1}-1}du_{1}=\sum_{m_{1}=0}^{\infty }\frac{{\left(b_{1}\right)}_{m_{1}}}{{\left(c\right)}_{m_{1}}}\frac{x_{1}^{m_{1}}}{m_{1}!}=\Phi_{2}^{\left(1\right)}\left(b_{1};c;x_{1}\right)={}_{1}F_{1}\left(b_{1},c;x_{1}\right) \end{array}$$
where ${}_{1}F_{1}\left(.\right)$ is the confluent hypergeometric function of the first kind (Section 9.21 in [35]).

Case n=2
(16)$$\begin{array}{cc} \; & \frac{1}{B(b_{1},b_{2},c-b_{1}-b_{2})}{\;\int \int \;}_{\begin{array}{c} u_{1}\geq 0,u_{2}\geq 0,\\ u_{1}+u_{2}\leq 1 \end{array}}u_{1}^{b_{1}-1}u_{2}^{b_{2}-1}e^{x_{1}u_{1}+x_{2}u_{2}}{\left(1-u_{1}-u_{2}\right)}^{c-b_{1}-b_{2}-1}du_{1}du_{2}\\ \; & =\sum_{m_{1}=0}^{\infty }\sum_{m_{2}=0}^{\infty }\frac{{\left(b_{1}\right)}_{m_{1}}{\left(b_{2}\right)}_{m_{2}}}{{\left(c\right)}_{m_{1}+m_{2}}}\frac{x_{1}^{m_{1}}}{m_{1}!}\frac{x_{2}^{m_{2}}}{m_{2}!}=\Phi_{2}^{\left(2\right)}\left(b_{1},b_{2};c;x_{1},x_{2}\right)=\Phi_{2}\left(b_{1},b_{2},c;x_{1},x_{2}\right) \end{array}$$
where the double series $\Phi_{2}$ is one of the components of the Humbert series of two variables [36] that generalize Kummer’s confluent hypergeometric series ${}_{1}F_{1}$ of one variable. The double series $\Phi_{2}$ converges absolutely at any $x_{1}$, $x_{2}\in C$.

### 3.2. Multiple Power Series $F_{N}^{\left(n\right)}$

**Definition** **2.**
*We define a new multiple power series, denoted by $F_{N}^{\left(n\right)}$ and given by*

(17)
$$\begin{array}{cc} \; & F_{N}^{\left(n\right)}\left(a;b_{1},\ldots ,b_{n};c,c_{n};x_{1},\ldots ,x_{n}\right)\\ \; & =x_{n}^{-a}\sum_{\begin{array}{c} m_{1},\ldots ,\\ m_{n}=0 \end{array}}^{+\infty }\frac{{\left(a\right)}_{\sum_{i=1}^{n}m_{i}}{\left(a-c_{n}+1\right)}_{\sum_{i=1}^{n}m_{i}}}{{\left(a+b_{n}-c_{n}+1\right)}_{\sum_{i=1}^{n}m_{i}}}\frac{\prod_{i=1}^{n-1}{\left(b_{i}\right)}_{m_{i}}}{{\left(c\right)}_{\sum_{i=1}^{n-1}m_{i}}}\prod_{i=1}^{n-1}{\left(\frac{x_{i}}{x_{n}}\right)}^{m_{i}}\frac{1}{m_{i}!}\frac{{\left(1-x_{n}^{-1}\right)}^{m_{n}}}{m_{n}!}. \end{array}$$

*The multiple power series (Equation 17) is absolutely convergent on the region $|x_{i}x_{n}^{-1}|+|1-x_{n}^{-1}|<1$ in $C^{n},\;\forall \;i\in \left\{1,\ldots ,n-1\right\}$.*


The multiple power series $F_{N}^{\left(n\right)}\left(.\right)$ can also be transformed into two other expressions as follows
(18)$$\begin{array}{c} F_{N}^{\left(n\right)}\left(a;b_{1},\ldots ,b_{n};c,c_{n};x_{1},\ldots ,x_{n}\right)\\ =\sum_{\begin{array}{c} m_{1},\ldots ,\\ m_{n}=0 \end{array}}^{+\infty }\frac{{\left(a-c_{n}+1\right)}_{\sum_{i=1}^{n-1}m_{i}}{\left(b_{n}\right)}_{m_{n}}{\left(a\right)}_{\sum_{i=1}^{n}m_{i}}}{{\left(a+b_{n}-c_{n}+1\right)}_{\sum_{i=1}^{n}m_{i}}}\frac{\prod_{i=1}^{n-1}{\left(b_{i}\right)}_{m_{i}}}{{\left(c\right)}_{\sum_{i=1}^{n-1}m_{i}}}\prod_{i=1}^{n-1}\frac{x_{i}^{m_{i}}}{m_{i}!}\frac{{\left(1-x_{n}\right)}^{m_{n}}}{m_{n}!}, \end{array}$$
(19)$$\begin{array}{cc} \; & =x_{n}^{1-c_{n}}\sum_{\begin{array}{c} m_{1},\ldots ,\\ m_{n}=0 \end{array}}^{+\infty }\frac{{\left(a-c_{n}+1\right)}_{\sum_{i=1}^{n}m_{i}}{\left(b_{n}-c_{n}+1\right)}_{m_{n}}{\left(a\right)}_{\sum_{i=1}^{n-1}m_{i}}}{{\left(a+b_{n}-c_{n}+1\right)}_{\sum_{i=1}^{n}m_{i}}}\frac{\prod_{i=1}^{n-1}{\left(b_{i}\right)}_{m_{i}}}{{\left(c\right)}_{\sum_{i=1}^{n-1}m_{i}}}\prod_{i=1}^{n-1}\frac{x_{i}^{m_{i}}}{m_{i}!}\frac{{\left(1-x_{n}\right)}^{m_{n}}}{m_{n}!}. \end{array}$$
By Horn’s rule for the determination of the convergence region (see [37], Section 5.7.2), the multiple power series (Equation 18) and (19) are absolutely convergent on region $|x_{i}|<1,\;\forall \;i\in \left\{1,\ldots ,n-1\right\},\;|1-x_{n}|<1$ in $C^{n}$.

Equation (Equation 18) can then be deduced from (Equation 17) by using the following development where the $F_{N}^{\left(p\right)}$ function can be written as
(20)$$\begin{array}{cc} \; & F_{N}^{\left(n\right)}\left(a;b_{1},\ldots ,b_{n};c,c_{n};x_{1},\ldots ,x_{n}\right)=x_{n}^{-a}\sum_{\begin{array}{c} m_{1},\ldots ,\\ m_{n-1}=0 \end{array}}^{+\infty }\frac{{\left(a\right)}_{\sum_{i=1}^{n-1}m_{i}}{\left(a-c_{n}+1\right)}_{\sum_{i=1}^{n-1}m_{i}}}{{\left(a+b_{n}-c_{n}+1\right)}_{\sum_{i=1}^{n-1}m_{i}}}\frac{\prod_{i=1}^{n-1}{\left(b_{i}\right)}_{m_{i}}}{{\left(c\right)}_{\sum_{i=1}^{n-1}m_{i}}}\\ \; & \times \prod_{i=1}^{n-1}{\left(\frac{x_{i}}{x_{n}}\right)}^{m_{i}}\frac{1}{m_{i}!}\sum_{m_{n}=0}^{\infty }\frac{{\left(\alpha \right)}_{m_{n}}{\left(\alpha -c_{n}+1\right)}_{m_{n}}}{{\left(\alpha +b_{n}-c_{n}+1\right)}_{m_{n}}}\frac{{\left(1-x_{n}^{-1}\right)}^{m_{n}}}{m_{n}!} \end{array}$$
and $\alpha =a+\sum_{i=1}^{n-1}m_{i}$ is used here to alleviate writing equations. Using the definition of Gauss’ hypergeometric series ${\;}_{2}F_{1}\left(.\right)$ [34] and the Pfaff transformation [38], we can write
(21)$$\begin{array}{cc} \sum_{m_{n}=0}^{\infty }\frac{{\left(\alpha \right)}_{m_{n}}{\left(\alpha -c_{n}+1\right)}_{m_{n}}}{{\left(\alpha +b_{n}-c_{n}+1\right)}_{m_{n}}}\frac{{\left(1-x_{n}^{-1}\right)}^{m_{n}}}{m_{n}!}\; & ={\;}_{2}F_{1}\left(\alpha ,\alpha -c_{n}+1;\alpha +b_{n}-c_{n}+1;1-x_{n}^{-1}\right) \end{array}$$
(22)$$\begin{array}{cc} \; & =x_{n}^{\alpha }{\;}_{2}F_{1}\left(\alpha ,b_{n};\alpha +b_{n}-c_{n}+1;1-x_{n}\right) \end{array}$$
(23)$$\begin{array}{cc} \; & =x_{n}^{\alpha }\sum_{m_{p}=0}^{\infty }\frac{{\left(\alpha \right)}_{m_{n}}{\left(b_{n}\right)}_{m_{n}}}{{\left(\alpha +b_{n}-c_{n}+1\right)}_{m_{n}}}\frac{{\left(1-x_{n}\right)}^{m_{n}}}{m_{n}!}. \end{array}$$
By substituting (Equation 23) into (Equation 20), and using the following two relations:(24)$$\begin{array}{cc} \; & {\left(a\right)}_{\sum_{i=1}^{n-1}m_{i}}{\left(\alpha \right)}_{m_{n}}={\left(a\right)}_{\sum_{i=1}^{n}m_{i}}\;, \end{array}$$
(25)$$\begin{array}{cc} \; & {\left(a+b_{n}-c_{n}+1\right)}_{\sum_{i=1}^{n-1}m_{i}}{\left(\alpha +b_{n}-c_{n}+1\right)}_{m_{n}}={\left(a+b_{n}-c_{n}+1\right)}_{\sum_{i=1}^{n}m_{i}} \end{array}$$
we can get (Equation 18).

The second transformation is given as follows
(26)$$\begin{array}{c} {\;}_{2}F_{1}\left(\alpha ,\alpha -c_{n}+1;b_{n}-c_{n}+\alpha +1;1-x_{n}^{-1}\right)\\ =x_{n}^{\alpha -c_{n}+1}{\;}_{2}F_{1}\left(b_{n}-c_{n}+1,\alpha -c_{n}+1;\alpha +b_{n}-c_{n}+1;1-x_{n}\right) \end{array}$$
(27)$$\begin{array}{cc} \; & =x_{n}^{\alpha -c_{n}+1}\sum_{m_{n}=0}^{\infty }\frac{{\left(\alpha -c_{n}+1\right)}_{m_{n}}{\left(b_{n}-c_{n}+1\right)}_{m_{n}}}{{\left(\alpha +b_{n}-c_{n}+1\right)}_{m_{n}}}\frac{{\left(1-x_{n}\right)}^{m_{n}}}{m_{n}!}. \end{array}$$
By substituting (Equation 27) into (Equation 20), we get (19).

**Lemma** **1.**
*The multiple power series $F_{N}^{\left(n\right)}$ is equal to the Lauricella D-hypergeometric function $F_{D}^{\left(n\right)}$ (see Appendix A) [39] when $a-c_{n}+1=c$ and is given as follows*

(28)
$$\begin{array}{cc} F_{N}^{\left(n\right)}\left(a;b_{1},\ldots ,b_{n};c,c_{n};x_{1},\ldots ,x_{n}\right)\; & =\sum_{\begin{array}{c} m_{1},\ldots ,\\ m_{n}=0 \end{array}}^{+\infty }\frac{{\left(a\right)}_{\sum_{i=1}^{n}m_{i}}\prod_{i=1}^{n}{\left(b_{i}\right)}_{m_{i}}}{{\left(a+b_{n}-c_{n}+1\right)}_{\sum_{i=1}^{n}m_{i}}}\prod_{i=1}^{n-1}\frac{x_{i}^{m_{i}}}{m_{i}!}\frac{{\left(1-x_{n}\right)}^{m_{n}}}{m_{n}!} \end{array}$$


(29)
$$\begin{array}{cc} \; & =F_{D}^{\left(n\right)}\left(a,b_{1},\ldots ,b_{n};a+b_{n}-c_{n}+1;x_{1},\ldots ,x_{n-1},1-x_{n}\right) \end{array}$$



**Proof.** By using Equation (Equation 18) of the multiple power series $F_{N}^{\left(n\right)}$ and after having simplified ${\left(a-c_{n}+1\right)}_{\sum_{i=1}^{n-1}m_{i}}$ to the numerator and ${\left(c\right)}_{\sum_{i=1}^{n-1}m_{i}}$ to the denominator, we can get the result. □

### 3.3. Integral Representation for $F_{N}^{\left(n+1\right)}$

**Proposition** **2.**
*The following integral representation is true for $\mathrm{Real}\left\{a\right\}>0,\mathrm{Real}\left\{a-c_{n+1}+1\right\}>0,\mathrm{and}\;\mathrm{Real}\left\{a-c_{n+1}+b_{n+1}+1\right\}>0$*

(30)
$$\begin{array}{cc} \; & \frac{\Gamma \left(a\right)\Gamma \left(a-c_{n+1}+1\right)}{\Gamma (a-c_{n+1}+b_{n+1}+1)}F_{N}^{\left(n+1\right)}\left(a;b_{1},\ldots ,b_{n+1};c,c_{n+1};x_{1},\ldots ,x_{n+1}\right)\\ \; & =\int_{0}^{\infty }e^{-r}r^{a-1}\Phi_{2}^{\left(n\right)}\left(b_{1},\ldots ,b_{n};c;rx_{1},\ldots ,rx_{n}\right)U\left(b_{n+1},c_{n+1};rx_{n+1}\right)dr \end{array}$$

*where U(·) is the confluent hypergeometric function of the second kind (Section 9.21 in [35]) defined for Real{b}>0, Real{z}>0 by the following integral representation*

(31)
$$\begin{array}{c} U\left(b,c;z\right)=\frac{1}{\Gamma (b)}\int_{0}^{\infty }e^{-zt}t^{b-1}{\left(1+t\right)}^{c-b-1}dt \end{array}$$

*and $\Phi_{2}^{\left(n\right)}\left(\cdot \right)$ is defined by Equation (Equation 6).*


**Proof.** The multiple power series $\Phi_{2}^{\left(n\right)}$ and the confluent hypergeometric function U(·) are absolutely convergent on [0,+∞]. Using these functions in the above integral and changing the order of integration and summation, which is easily justified by absolute convergence, we get
(32)$$\begin{array}{cc} \; & \int_{0}^{\infty }e^{-r}r^{a-1}\Phi_{2}^{\left(n\right)}\left(b_{1},\ldots ,b_{n};c;rx_{1},\ldots ,rx_{n}\right)U\left(b_{n+1},c_{n+1};rx_{n+1}\right)dr\\ \; & =\sum_{m_{1}=0}^{\infty }..\sum_{m_{n}=0}^{\infty }\frac{{\left(b_{1}\right)}_{m_{1}}\ldots {\left(b_{n}\right)}_{m_{n}}}{{\left(c\right)}_{\sum_{i=1}^{n}m_{i}}}\left(\prod_{i=1}^{n}\frac{x_{i}^{m_{i}}}{m_{i}!}\right)I \end{array}$$
where integral I is defined as follows
(33)$$\begin{array}{cc} I & =\int_{0}^{\infty }e^{-r}r^{a-1+\sum_{i=1}^{n}m_{i}}U\left(b_{n+1},c_{n+1};rx_{n+1}\right)dr. \end{array}$$
Substituting the integral expression of U(·) in the previous equation and replacing $\alpha =a+\sum_{i=1}^{n}m_{i}$ to alleviate writing equations, we have
(34)$$\begin{array}{cc} I & =\frac{1}{\Gamma (b_{n+1})}\int_{0}^{\infty }\int_{0}^{\infty }\frac{e^{-(1+x_{n+1}t)r}r^{\alpha -1}t^{b_{n+1}-1}}{{\left(1+t\right)}^{-(c_{n+1}-b_{n+1}-1)}}drdt. \end{array}$$
Knowing that [35]
(35)$$\begin{array}{cc} \; & \int_{0}^{\infty }e^{-(1+x_{n+1}t)r}r^{\alpha -1}dr=\frac{\Gamma (\alpha )}{{\left(1+x_{n+1}t\right)}^{\alpha }} \end{array}$$
and
(36)$$\begin{array}{cc} \; & \int_{0}^{\infty }\frac{t^{b_{n+1}-1}{\left(1+t\right)}^{c_{n+1}-b_{n+1}-1}}{{\left(1+x_{n+1}t\right)}^{\alpha }}dt=\frac{\Gamma \left(b_{n+1}\right)\Gamma \left(\alpha -c_{n+1}+1\right)}{\Gamma (\alpha +b_{n+1}-c_{n+1}+1)}{\;}_{2}F_{1}\left(\alpha ,b_{n+1};\alpha +b_{n+1}-c_{n+1}+1;1-x_{n+1}\right) \end{array}$$
the new expression of I is then given by
(37)$$\begin{array}{cc} I & =\frac{\Gamma \left(\alpha \right)\Gamma \left(\alpha -c_{n+1}+1\right)}{\Gamma (\alpha +b_{n+1}-c_{n+1}+1)}\sum_{m_{n+1}=0}^{+\infty }\frac{{\left(\alpha \right)}_{m_{n+1}}{\left(b_{n+1}\right)}_{m_{n+1}}}{{\left(\alpha +b_{n+1}-c_{n+1}+1\right)}_{m_{n+1}}}\frac{{\left(1-x_{n+1}\right)}^{m_{n+1}}}{m_{n+1}!}. \end{array}$$
Using the fact that $\Gamma \left(\alpha \right)=\Gamma \left(a\right){\left(a\right)}_{\sum_{i=1}^{n}m_{i}}$ and ${\left(a\right)}_{\sum_{i=1}^{n}m_{i}}{\left(\alpha \right)}_{m_{n+1}}={\left(a\right)}_{\sum_{i=1}^{n+1}m_{i}}$, and developing the same method to $\Gamma (\alpha +b_{n+1}-c_{n+1}+1)$, the final complete expression of the integral is then given by
(38)$$\begin{array}{cc} \; & \frac{\Gamma \left(a\right)\Gamma \left(a-c_{n+1}+1\right)}{\Gamma (a+b_{n+1}-c_{n+1}+1)}\sum_{m_{1}=0}^{\infty }..\sum_{m_{n+1}=0}^{\infty }\frac{{\left(b_{1}\right)}_{m_{1}}\ldots {\left(b_{n}\right)}_{m_{n}}}{{\left(c\right)}_{\sum_{i=1}^{n}m_{i}}}\frac{{\left(a-c_{n+1}+1\right)}_{\sum_{i=1}^{n}m_{i}}{\left(b_{n+1}\right)}_{m_{n+1}}{\left(a\right)}_{\sum_{i=1}^{n+1}m_{i}}}{{\left(a+b_{n+1}-c_{n+1}+1\right)}_{\sum_{i=1}^{n+1}m_{i}}}\prod_{i=1}^{n}\frac{x_{i}^{m_{i}}}{m_{i}!}\\ \; & \times \frac{{\left(1-x_{n+1}\right)}^{m_{n+1}}}{m_{n+1}!}=\frac{\Gamma \left(a\right)\Gamma \left(a-c_{n+1}+1\right)}{\Gamma (a-c_{n+1}+b_{n+1}+1)}F_{N}^{\left(n+1\right)}\left(a;b_{1},\ldots ,b_{n+1};c,c_{n+1};x_{1},\ldots ,x_{n+1}\right). \end{array}$$
□

## 4. Expression of $E_{X^{1}}\left\{\ln\left[1+X^{T}\Sigma_{1}^{-1}X\right]\right\}$


**Proposition** **3.**
*Let $X^{1}$ be a random vector that follows a central MCD with pdf given by $f_{X^{1}}\left(x|\Sigma_{1},p\right)$. Expectation $E_{X^{1}}\left\{\ln\left[1+X^{T}\Sigma_{1}^{-1}X\right]\right\}$ is given as follows*

(39)
$$\begin{array}{c} E_{X^{1}}\left\{\ln[1+X^{T}\Sigma_{1}^{-1}X]\right\}=\psi \left(\frac{1+p}{2}\right)-\psi \left(\frac{1}{2}\right) \end{array}$$

*where ψ(.) is the digamma function defined as the logarithmic derivative of the Gamma function (Section 8.36 in [35]).*


**Proof.** Expectation $E_{X^{1}}\left\{\ln\left[1+X^{T}\Sigma_{1}^{-1}X\right]\right\}$ is developed as follows
(40)$$\begin{array}{cc} \; & E_{X^{1}}\left\{\ln\left[1+X^{T}\Sigma_{1}^{-1}X\right]\right\}=\frac{A}{|\Sigma_{1}{|}^{\frac{1}{2}}}\int_{R^{p}}\frac{\ln[1+x^{T}\Sigma_{1}^{-1}x]}{{\left[1+x^{T}\Sigma_{1}^{-1}x\right]}^{\frac{1+p}{2}}}dx \end{array}$$
where $A=\Gamma (\frac{1+p}{2})\pi^{-\frac{1+p}{2}}$. Utilizing the following property $\int \log\left(x\right)f\left(x\right)dx=\frac{\partial }{\partial a}\int x^{a}f\left(x\right)$ $dx{|}_{a=0}$, as a consequence the expectation is given as follows
(41)$$\begin{array}{cc} \; & E_{X^{1}}\left\{\ln\left[1+X^{T}\Sigma_{1}^{-1}X\right]\right\}=\frac{A}{|\Sigma_{1}{|}^{\frac{1}{2}}}\frac{\partial }{\partial a}\int_{R^{p}}{\left[1+x^{T}\Sigma_{1}^{-1}x\right]}^{a-\frac{1+p}{2}}dx{|}_{a=0} \end{array}$$Consider the transformation $y=\Sigma_{1}^{-1/2}x$ where $y={\left[y_{1},y_{2},\ldots ,y_{p}\right]}^{T}$. The Jacobian determinant is given by $dy=|\Sigma_{1}{|}^{-1/2}dx$ (Theorem 1.12 in [40]). The new expression of the expectation is given by
(42)$$\begin{array}{cc} \; & E_{X^{1}}\left\{\ln\left[1+X^{T}\Sigma_{1}^{-1}X\right]\right\}=A\frac{\partial }{\partial a}\int_{R^{p}}{\left[1+y^{T}y\right]}^{a-\frac{1+p}{2}}dy{|}_{a=0}. \end{array}$$
Let $u=y^{T}y$ be a transformation where the Jacobian determinant is given by (Lemma 13.3.1 in [41])
(43)$$\begin{array}{cc} \; & dy=\frac{\pi^{\frac{p}{2}}}{\Gamma (\frac{p}{2})}u^{\frac{p}{2}-1}du. \end{array}$$
The new expectation is as follows
(44)$$\begin{array}{cc} \; & E_{X^{1}}\left\{\ln\left[1+X^{T}\Sigma_{1}^{-1}X\right]\right\}=\frac{\Gamma (\frac{1+p}{2})}{\pi^{1/2}\Gamma \left(\frac{p}{2}\right)}\frac{\partial }{\partial a}\int_{0}^{+\infty }u^{\frac{p}{2}-1}{\left(1+u\right)}^{a-\frac{1+p}{2}}du{|}_{a=0} \end{array}$$
Using the definition of beta function, we can write that
(45)$$\begin{array}{cc} \; & \int_{0}^{+\infty }u^{\frac{p}{2}-1}{\left(1+u\right)}^{a-\frac{1+p}{2}}du=\frac{\Gamma \left(\frac{p}{2}\right)\Gamma \left(\frac{1}{2}-a\right)}{\Gamma (\frac{1+p}{2}-a)}. \end{array}$$
The derivative of the last integral w.r.t *a* is given by
(46)$$\begin{array}{cc} \; & \frac{\partial }{\partial a}\int_{0}^{+\infty }u^{\frac{p}{2}-1}{\left(1+u\right)}^{a-\frac{1+p}{2}}du{|}_{a=0}=\frac{\Gamma \left(\frac{p}{2}\right)\Gamma \left(\frac{1}{2}\right)}{\Gamma (\frac{1+p}{2})}\left[\psi \left(\frac{1+p}{2}\right)-\psi \left(\frac{1}{2}\right)\right] \end{array}$$
Finally, the expression of $E_{X^{1}}\left\{\ln\left[1+X^{T}\Sigma_{1}^{-1}X\right]\right\}$ is given by
(47)$$\begin{array}{cc} \; & E_{X^{1}}\left\{\ln\left[1+X^{T}\Sigma_{1}^{-1}X\right]\right\}=\psi \left(\frac{1+p}{2}\right)-\psi \left(\frac{1}{2}\right). \end{array}$$
□

## 5. Expression of $E_{X^{1}}\left\{\ln\left[1+X^{T}\Sigma_{2}^{-1}X\right]\right\}$


**Proposition** **4.**
*Let $X^{1}$ and $X^{2}$ be two random vectors that follow central MCDs with pdfs given, respectively, by $f_{X^{1}}\left(x|\Sigma_{1},p\right)$ and $f_{X^{2}}\left(x|\Sigma_{2},p\right)$. Expectation $E_{X^{1}}\left\{\ln\left[1+X^{T}\Sigma_{2}^{-1}X\right]\right\}$ is given as follows*

(48)
$$\begin{array}{cc} \; & E_{X^{1}}\left\{\ln\left[1+X^{T}\Sigma_{2}^{-1}X\right]\right\}=\psi \left(\frac{1+p}{2}\right)-\psi \left(\frac{1}{2}\right)+\ln\lambda_{p}\\ \; & -\frac{\partial }{\partial a}\left\{F_{D}^{\left(p\right)}\left(a,\underset{p}{\underbrace{\frac{1}{2},\frac{1}{2},\ldots ,\frac{1}{2},a+\frac{1}{2}}};a+\frac{1+p}{2};1-\frac{\lambda_{1}}{\lambda_{p}},\ldots ,1-\frac{\lambda_{p-1}}{\lambda_{p}},1-\frac{1}{\lambda_{p}}\right)\right\}{|}_{a=0}. \end{array}$$

*where $\lambda_{1}$,…, $\lambda_{p}$ are the eigenvalues of the real matrix $\Sigma_{1}\Sigma_{2}^{-1}$, and $F_{D}^{\left(p\right)}\left(.\right)$ represents the Lauricella D-hypergeometric function defined for p variables.*


**Proof.** To prove Proposition 4, different steps are necessary. They are described in the following:

### 5.1. First Step: Eigenvalue Expression

Expectation $E_{X^{1}}\left\{\ln\left[1+X^{T}\Sigma_{2}^{-1}X\right]\right\}$ is computed as follows
(49)$$\begin{array}{cc} \;E_{X^{1}}\left\{\ln\left[1+X^{T}\Sigma_{2}^{-1}X\right]\right\} & =\frac{A}{|\Sigma_{1}{|}^{\frac{1}{2}}}\int_{R^{p}}\frac{\ln[1+x^{T}\Sigma_{2}^{-1}x]}{{\left[1+x^{T}\Sigma_{1}^{-1}x\right]}^{\frac{1+p}{2}}}dx \end{array}$$
where $A=\Gamma (\frac{1+p}{2})\pi^{-\frac{1+p}{2}}$. Consider transformation $y=\Sigma_{1}^{-1/2}x$ where $y={\left[y_{1},y_{2},\ldots ,y_{p}\right]}^{T}$. The Jacobian determinant is given by $dy=|\Sigma_{1}{|}^{-1/2}dx$ (Theorem 1.12 in [40]) and matrix $\Sigma =\Sigma_{1}^{\frac{1}{2}}\Sigma_{2}^{-1}\Sigma_{1}^{\frac{1}{2}}$ is a real symmetric matrix since $\Sigma_{1}$ and $\Sigma_{2}$ are real symmetric matrixes. Then, the expectation is evaluated as follows
(50)$$\begin{array}{cc} E_{X^{1}}\left\{\ln\left[1+X^{T}\Sigma_{2}^{-1}X\right]\right\} & =A\int_{R^{p}}\frac{\ln[1+y^{T}\Sigma y]}{{\left[1+y^{T}y\right]}^{\frac{1+p}{2}}}dy. \end{array}$$
Matrix Σ can be diagonalized by an orthogonal matrix P with $P^{-1}=P^{T}$ and $\Sigma =PDP^{-1}$ where D is a diagonal matrix composed of the eigenvalues of Σ. Considering that $y^{T}\Sigma y=\mathrm{tr}\left(\Sigma yy^{T}\right)=\mathrm{tr}\left(PDP^{T}yy^{T}\right)=\mathrm{tr}\left(DP^{T}yy^{T}P\right)$, the expectation can be written as
(51)$$\begin{array}{cc} E_{X^{1}}\left\{\ln\left[1+X^{T}\Sigma_{2}^{-1}X\right]\right\} & =A\int_{R^{p}}\frac{\ln[1+\mathrm{tr}\left(DP^{T}yy^{T}P\right)]}{{\left[1+y^{T}y\right]}^{\frac{1+p}{2}}}dy. \end{array}$$
Let $z=P^{T}y$ with $z={\left[z_{1},z_{2},\ldots ,z_{p}\right]}^{T}$ be a transformation where the Jacobian determinant is given by $dz=|P^{T}|dy=dy$. Using the fact that $\mathrm{tr}\left(DP^{T}yy^{T}P\right)=\mathrm{tr}\left(Dzz^{T}\right)=z^{T}Dz$ and $y^{T}y=z^{T}P^{T}Pz=z^{T}z$, then the previous expectation (Equation 51) is given as follows
(52)$$\begin{array}{cc} E_{X^{1}}\left\{\ln\left[1+X^{T}\Sigma_{2}^{-1}X\right]\right\} & =A\int_{R^{p}}\frac{\ln[1+z^{T}Dz]}{{\left[1+z^{T}z\right]}^{\frac{1+p}{2}}}dz \end{array}$$
(53)$$\begin{array}{cc} \; & =A\int_{R}..\int_{R}\frac{\ln[1+\sum_{i=1}^{p}\lambda_{i}z_{i}^{2}]}{{\left[1+\sum_{i=1}^{p}z_{i}^{2}\right]}^{\frac{1+p}{2}}}dz_{1}\ldots dz_{p} \end{array}$$
where $\lambda_{1}$,…, $\lambda_{p}$ are the eigenvalues of $\Sigma_{1}\Sigma_{2}^{-1}$.

### 5.2. Second Step: Polar Decomposition

Let the independent real variables $z_{1},\ldots ,z_{p}$ be transformed to the general polar coordinates *r*, $\theta_{1},\ldots ,\theta_{p-1}$ as follows, where r>0, $-\pi /2<\theta_{j}\leq \pi /2$, j=1,…,p−2, $-\pi <\theta_{p-1}\leq \pi$ [40],
(54)$$\begin{array}{cc} z_{1} & =r\sin\theta_{1} \end{array}$$
(55)$$\begin{array}{cc} z_{2} & =r\cos\theta_{1}\sin\theta_{2} \end{array}$$
(56)$$\begin{array}{cc} z_{j} & =r\cos\theta_{1}\cos\theta_{2}\ldots \cos\theta_{j-1}\sin\theta_{j},\;j=2,3,\ldots ,p-1 \end{array}$$
(57)$$\begin{array}{cc} z_{p} & =r\cos\theta_{1}\cos\theta_{2}\ldots \cos\theta_{p-1}. \end{array}$$
The Jacobian determinant according to theorem (1.24) in [40] is
(58)$$\begin{array}{c} dz_{1}\ldots dz_{p}=r^{p-1}\prod_{j=1}^{p-1}{\left|\cos\theta_{j}\right|}^{p-j-1}drd\theta_{j}. \end{array}$$
It is clear that with the last transformations, we get $\sum_{i=1}^{p}z_{i}^{2}=r^{2}$ and the multiple integral in (Equation 53) is then given as follows
(59)$$\begin{array}{cc} \; & E_{X^{1}}\left\{\ln\left[1+X^{T}\Sigma_{2}^{-1}X\right]\right\}=A\int_{0}^{+\infty }\frac{r^{p-1}}{{\left[1+r^{2}\right]}^{\frac{1+p}{2}}}\int_{-\pi /2}^{\pi /2}\;..\int_{-\pi }^{\pi }\left(\prod_{j=1}^{p-1}{\left|\cos\theta_{j}\right|}^{p-j-1}\right)\times \\ \; & \ln\left[1+r^{2}\left(\lambda_{1}\sin^{2}\theta_{1}+\ldots +\lambda_{p}\cos^{2}\theta_{1}\ldots \cos^{2}\theta_{p-1}\right)\right]dr\prod_{j=1}^{p-1}d\theta_{j}. \end{array}$$
By replacing the expression of $\sin^{2}\theta_{j}$ by $1-\cos^{2}\theta_{j}$, for j=1,…,p−1, we have the following expression
(60)$$\begin{array}{cc} \; & \lambda_{1}\sin^{2}\theta_{1}+\ldots +\lambda_{p}\cos^{2}\theta_{1}\ldots \cos^{2}\theta_{p-1}=\lambda_{1}+\left(\lambda_{2}-\lambda_{1}\right)\cos^{2}\theta_{1}\\ \; & +\ldots +\left(\lambda_{p}-\lambda_{p-1}\right)\cos^{2}\theta_{1}\cos^{2}\theta_{2}\ldots \cos^{2}\theta_{p-1}. \end{array}$$
Let $x_{i}=\cos^{2}\theta_{i}$ be a transformation to use where $dx_{i}=2x_{i}^{1/2}{\left(1-x_{i}\right)}^{1/2}d\theta_{i}$. Then the expectation given by the multiple integral over all $\theta_{j}$, j=1,…,p−1 is as follows
(61)$$\begin{array}{c} 2A\int_{0}^{+\infty }\frac{r^{p-1}}{{\left[1+r^{2}\right]}^{\frac{1+p}{2}}}\int_{0}^{1}\;\ldots \;\int_{0}^{1}\left(\prod_{j=1}^{p-1}x_{j}^{\frac{p-j}{2}-1}{\left(1-x_{j}\right)}^{-\frac{1}{2}}\right)\ln\left[1+r^{2}B_{p}\left(x_{1},\ldots ,x_{p-1}\right)\right]drdx_{1}\ldots dx_{p-1} \end{array}$$
where $B_{p}\left(x_{1},\ldots ,x_{p-1}\right)=\lambda_{1}+\left(\lambda_{2}-\lambda_{1}\right)x_{1}+\ldots +\left(\lambda_{p}-\lambda_{p-1}\right)x_{1}x_{2}\ldots x_{p-1}$, p≥1 and $B_{1}=\lambda_{1}$. In the following, we use the notation $B_{p}$ instead of $B_{p}\left(x_{1},\ldots ,x_{p-1}\right)$ to alleviate writing equations.

Let $t=r^{2}$ be transformation to use. Then, one can write
(62)$$\begin{array}{cc} \; & =A\int_{0}^{+\infty }\frac{t^{\frac{p}{2}-1}}{{\left[1+t\right]}^{\frac{1+p}{2}}}\int_{0}^{1}\;\ldots \;\int_{0}^{1}\left(\prod_{j=1}^{p-1}x_{j}^{\frac{p-j}{2}-1}{\left(1-x_{j}\right)}^{-\frac{1}{2}}\right)\ln\left[1+tB_{p}\right]dtdx_{1}\ldots dx_{p-1}. \end{array}$$
In order to solve the integral in (Equation 62), we consider the following property given by ∫log(x)f(x) $dx=-\frac{\partial }{\partial a}\int x^{-a}f\left(x\right)dx{|}_{a=0}$ and the following equation given as follows
(63)$$\begin{array}{c} {\left(1+B_{p}t\right)}^{-a}=\frac{1}{\Gamma (a)}\int_{0}^{+\infty }y^{a-1}e^{-(1+B_{p}t)y}dy. \end{array}$$
Making use of the above equation, we obtain a new expression of (Equation 62) given as follows
(64)$$\begin{array}{c} E_{X^{1}}\left\{\ln\left[1+X^{T}\Sigma_{2}^{-1}X\right]\right\}\\ =-\frac{\partial }{\partial a}\left\{\frac{A}{\Gamma (a)}\int_{0}^{+\infty }\frac{t^{\frac{p}{2}-1}}{{\left[1+t\right]}^{\frac{1+p}{2}}}\int_{0}^{+\infty }y^{a-1}e^{-(1+B_{p}t)y}\int_{0}^{1}\ldots \int_{0}^{1}\prod_{j=1}^{p-1}x_{j}^{\frac{p-j}{2}-1}{\left(1-x_{j}\right)}^{-\frac{1}{2}}dx_{j}dydt\right\}{|}_{a=0} \end{array}$$
(65)$$\begin{array}{cc} \; & =-\frac{\partial }{\partial a}\left\{\frac{A}{\Gamma (a)}\int_{0}^{+\infty }\frac{t^{\frac{p}{2}-1}}{{\left[1+t\right]}^{\frac{1+p}{2}}}\int_{0}^{+\infty }y^{a-1}e^{-y}H\left(t,y\right)dydt\right\}{|}_{a=0} \end{array}$$
where H(t,y) is defined as
(66)$$\begin{array}{cc} \; & H\left(t,y\right)=\int_{0}^{1}\;\ldots \;\int_{0}^{1}e^{-B_{p}ty}\prod_{j=1}^{p-1}x_{j}^{\frac{p-j}{2}-1}{\left(1-x_{j}\right)}^{-\frac{1}{2}}dx_{j}. \end{array}$$

### 5.3. Third Step: Expression for H(t,y) by Humbert and Beta Functions

Let $x_{i}^{\prime }=1-x_{i}$, i=1,…,p−1 be transformations to use. Then
(67)$$\begin{array}{cc} \left(\lambda_{2}-\lambda_{1}\right)x_{1} & =\left(\lambda_{2}-\lambda_{1}\right)\left(1-x_{1}^{\prime }\right) \end{array}$$
(68)$$\begin{array}{cc} \left(\lambda_{3}-\lambda_{2}\right)x_{1}x_{2} & =\left(\lambda_{3}-\lambda_{2}\right)\left(1-x_{1}^{\prime }\right)\left[1-x_{2}^{\prime }\right] \end{array}$$
(69)$$\begin{array}{cc} \left(\lambda_{4}-\lambda_{3}\right)x_{1}x_{2}x_{3} & =\left(\lambda_{4}-\lambda_{3}\right)\left(1-x_{1}^{\prime }\right)\left(1-x_{2}^{\prime }\right)\left[1-x_{3}^{\prime }\right]\\ \vdots & =\vdots \end{array}$$
(70)$$\begin{array}{cc} \left(\lambda_{p}-\lambda_{p-1}\right)\prod_{i=1}^{p-1}x_{i} & =\left(\lambda_{p}-\lambda_{p-1}\right)\prod_{i=1}^{p-1}\left(1-x_{i}^{\prime }\right). \end{array}$$
Adding equations from (Equation 67) to (70), we can state that the new expression of the function $B_{p}$ becomes
(71)$$\begin{array}{cc} \; & B_{p}=\lambda_{p}-\left(\lambda_{p}-\lambda_{1}\right)x_{1}^{\prime }-\left(\lambda_{p}-\lambda_{2}\right)\left(1-x_{1}^{\prime }\right)x_{2}^{\prime }-\left(\lambda_{p}-\lambda_{3}\right)\left(1-x_{1}^{\prime }\right)\left(1-x_{2}^{\prime }\right)x_{3}^{\prime }\\ \; & -\ldots -\left(\lambda_{p}-\lambda_{p-1}\right)\left(1-x_{1}^{\prime }\right)\ldots \left(1-x_{p-2}^{\prime }\right)x_{p-1}^{\prime }. \end{array}$$
Then, the multiple integral H(t,y) given by (Equation 66) can be written otherwise
(72)$$\begin{array}{cc} \; & H\left(t,y\right)=\int_{0}^{1}\;\ldots \;\int_{0}^{1}e^{-B_{p}ty}\prod_{j=1}^{p-1}{\left(1-x_{j}^{\prime }\right)}^{\frac{p-j}{2}-1}{x_{j}^{\prime }}^{-\frac{1}{2}}dx_{1}^{\prime }\ldots dx_{p-1}^{\prime }. \end{array}$$
Let the real variables $x_{1}^{\prime },x_{2}^{\prime },\ldots ,x_{p-1}^{\prime }$ be transformed to the real variables $u_{1},u_{2},\ldots ,u_{p-1}$ as follows
(73)$$\begin{array}{cc} \; & u_{1}=x_{1}^{\prime } \end{array}$$
(74)$$\begin{array}{cc} \; & u_{2}=\left(1-x_{1}^{\prime }\right)x_{2}^{\prime }=\left(1-u_{1}\right)x_{2}^{\prime } \end{array}$$
(75)$$\begin{array}{cc} \; & u_{3}=\left(1-x_{1}^{\prime }\right)\left(1-x_{2}^{\prime }\right)x_{3}^{\prime }=\left(1-u_{1}-u_{2}\right)x_{3}^{\prime }\\ \; & \vdots \end{array}$$
(76)$$\begin{array}{cc} \; & u_{p-1}=\prod_{i=1}^{p-2}\left(1-x_{i}^{\prime }\right)x_{p-1}^{\prime }=\left(1-\sum_{i=1}^{p-2}u_{i}\right)x_{p-1}^{\prime }. \end{array}$$
The Jacobian determinant is given by
(77)$$\begin{array}{c} du_{1}\ldots du_{p-1}=\prod_{j=1}^{p-1}\left(1-\sum_{i=1}^{j-1}u_{i}\right)dx_{1}^{\prime }\ldots dx_{p-1}^{\prime }. \end{array}$$
Accordingly, the new expression of $B_{p}$ becomes
(78)$$\begin{array}{c} B_{p}=\lambda_{p}-\sum_{i=1}^{p-1}\left(\lambda_{p}-\lambda_{i}\right)u_{i}. \end{array}$$
As a consequence, the new domain of the multiple integral (Equation 72) is $\Delta =\{\left(u_{1},u_{2},\ldots ,u_{p-1}\right)\in R^{p-1};0\leq u_{1}\leq 1,\;0\leq u_{2}\leq 1-u_{1},\;0\leq u_{3}\leq 1-u_{1}-u_{2},\ldots ,\mathrm{and}\;0\leq u_{p-1}\leq 1-u_{1}-u_{2}\ldots -u_{p-2}\}$, and the expression of H(t,y) is given as follows
(79)$$\begin{array}{cc} H\left(t,y\right) & =\int \;\ldots \;\int_{\Delta }e^{-B_{p}ty}\prod_{j=1}^{p-1}{\left(1-\sum_{i=1}^{j-1}u_{i}\right)}^{-1}{\left(\frac{u_{j}}{1-\sum_{i=1}^{j-1}u_{i}}\right)}^{-\frac{1}{2}}\prod_{j=1}^{p-1}{\left(1-\frac{u_{j}}{1-\sum_{i=1}^{j-1}u_{i}}\right)}^{\frac{p-j}{2}-1}du_{j} \end{array}$$
(80)$$\begin{array}{cc} \; & =\int \;\ldots \;\int_{\Delta }e^{-B_{p}ty}\prod_{j=1}^{p-1}u_{j}^{-\frac{1}{2}}{\left(1-\sum_{i=1}^{j}u_{i}\right)}^{\frac{p-j}{2}-1}{\left(1-\sum_{i=1}^{j-1}u_{i}\right)}^{\frac{1}{2}-\frac{p-j}{2}}du_{1}\ldots du_{p-1} \end{array}$$
(81)$$\begin{array}{cc} \; & =\int \;\ldots \;\int_{\Delta }e^{-B_{p}ty}{\left(1-\sum_{i=1}^{p-1}u_{i}\right)}^{\frac{p}{2}-\frac{p-1}{2}-1}\prod_{j=1}^{p-1}u_{j}^{-\frac{1}{2}}du_{j} \end{array}$$
(82)$$\begin{array}{cc} \; & =e^{-\lambda_{p}ty}\int \;\ldots \;\int_{\Delta }{\left(1-\sum_{i=1}^{p-1}u_{i}\right)}^{-\frac{1}{2}}\prod_{i=1}^{p-1}u_{i}^{-\frac{1}{2}}e^{\left(\lambda_{p}-\lambda_{i}\right)u_{i}ty}du_{i}. \end{array}$$
Using Proposition 1, we subsequently find that
(83)$$\begin{array}{cc} \; & H\left(t,y\right)=e^{-\lambda_{p}ty}B\left(\underset{p}{\underbrace{\frac{1}{2},\ldots ,\frac{1}{2}}}\right)\Phi_{2}^{\left(p-1\right)}\left(\underset{p-1}{\underbrace{\frac{1}{2},\ldots ,\frac{1}{2}}};\frac{p}{2};\left(\lambda_{p}-\lambda_{1}\right)ty,\left(\lambda_{p}-\lambda_{2}\right)ty,\ldots ,\left(\lambda_{p}-\lambda_{p-1}\right)ty\right). \end{array}$$
where $\Phi_{2}^{\left(p-1\right)}\left(.\right)$ is the Humbert series of p−1 variables and $B(\frac{1}{2},\ldots ,\frac{1}{2})$ is the multivariate beta function. Applying the following successive two transformations r=ty (dr=tdy) and u=1/t ($du=-u^{2}dt$), the new expression of the expectation given by (Equation 65) is written as follows
(84)$$\begin{array}{cc} \; & E_{X^{1}}\left\{\ln\left[1+X^{T}\Sigma_{2}^{-1}X\right]\right\}=-\frac{\partial }{\partial a}\{\frac{A}{\Gamma (a)}B\left(\underset{p}{\underbrace{\frac{1}{2},\ldots ,\frac{1}{2}}}\right)\int_{0}^{+\infty }r^{a-1}e^{-\lambda_{p}r}\\ \; & \times \Phi_{2}^{\left(p-1\right)}\left(\underset{p-1}{\underbrace{\frac{1}{2},\ldots ,\frac{1}{2}}};\frac{p}{2};\left(\lambda_{p}-\lambda_{1}\right)r,\ldots ,\left(\lambda_{p}-\lambda_{p-1}\right)r\right)\left(\int_{0}^{+\infty }u^{a-\frac{1}{2}}{\left(1+u\right)}^{-\frac{1+p}{2}}e^{-ru}du\right)dr\}{|}_{a=0}. \end{array}$$

### 5.4. Final Step

The last integral is related to the confluent hypergeometric function of the second kind U(.) as follows
(85)$$\begin{array}{c} \int_{0}^{+\infty }u^{a-\frac{1}{2}}{\left(1+u\right)}^{-\frac{1+p}{2}}e^{-ru}du=\Gamma \left(a+\frac{1}{2}\right)U\left(a+\frac{1}{2},a+1-\frac{p}{2},r\right). \end{array}$$
As a consequence, the new expression is
(86)$$\begin{array}{cc} \; & E_{X^{1}}\left\{\ln\left[1+X^{T}\Sigma_{2}^{-1}X\right]\right\}=-\frac{\partial }{\partial a}\{A\frac{\Gamma (a+\frac{1}{2})}{\Gamma (a)}B\left(\frac{1}{2},\ldots ,\frac{1}{2}\right)\\ \; & \times \int_{0}^{+\infty }r^{a-1}e^{-\lambda_{p}r}\Phi_{2}^{\left(p-1\right)}\left(\frac{1}{2},\ldots ,\frac{1}{2};\frac{p}{2};1;\left(\lambda_{p}-\lambda_{1}\right)r,\ldots ,\left(\lambda_{p}-\lambda_{p-1}\right)r\right)U\left(a+\frac{1}{2},a+1-\frac{p}{2},r\right)dr\}{|}_{a=0}. \end{array}$$
Using the transformation $r^{\prime }=\lambda_{p}r$ and the Proposition 2, and taking into account the expression of *A*, the new expression becomes
(87)$$\begin{array}{cc} \; & E_{X^{1}}\left\{\ln\left[1+X^{T}\Sigma_{2}^{-1}X\right]\right\}=-\frac{\partial }{\partial a}\{\frac{B(a+\frac{1}{2},\frac{p}{2})}{B(\frac{1}{2},\frac{p}{2})}\lambda_{p}^{-a}\\ \; & \times F_{N}^{\left(p\right)}\left(a;\underset{p}{\underbrace{\frac{1}{2},\ldots ,\frac{1}{2},a+\frac{1}{2}}};\frac{p}{2},a-\frac{p}{2}+1;1-\frac{\lambda_{1}}{\lambda_{p}},\ldots ,1-\frac{\lambda_{p-1}}{\lambda_{p}},\lambda_{p}^{-1}\right)\}{|}_{a=0} \end{array}$$
Knowing that
(88)$$\begin{array}{cc} \; & \frac{\partial }{\partial a}\left\{\frac{B(\frac{p}{2},a+\frac{1}{2})}{B(\frac{p}{2},\frac{1}{2})}\right\}{|}_{a=0}=\psi \left(\frac{1}{2}\right)-\psi \left(\frac{1+p}{2}\right),\;\mathrm{and} \end{array}$$
(89)$$\begin{array}{cc} \; & F_{N}^{\left(p\right)}\left(a;\frac{1}{2},\ldots ,\frac{1}{2},a+\frac{1}{2};\frac{p}{2},a-\frac{p}{2}+1;1-\frac{\lambda_{1}}{\lambda_{p}},\ldots ,1-\frac{\lambda_{p-1}}{\lambda_{p}},\lambda_{p}^{-1}\right){|}_{a=0}=1, \end{array}$$
the new expression of $E_{X^{1}}\left\{\ln\left[1+X^{T}\Sigma_{2}^{-1}X\right]\right\}$ becomes
(90)$$\begin{array}{cc} \; & E_{X^{1}}\left\{\ln\left[1+X^{T}\Sigma_{2}^{-1}X\right]\right\}=\psi \left(\frac{1+p}{2}\right)-\psi \left(\frac{1}{2}\right)\\ \; & -\frac{\partial }{\partial a}\left\{\lambda_{p}^{-a}F_{N}^{\left(p\right)}\left(a;\underset{p}{\underbrace{\frac{1}{2},\ldots ,\frac{1}{2},a+\frac{1}{2}}};\frac{p}{2},a-\frac{p}{2}+1;1-\frac{\lambda_{1}}{\lambda_{p}},\ldots ,1-\frac{\lambda_{p-1}}{\lambda_{p}},\lambda_{p}^{-1}\right)\right\}{|}_{a=0}. \end{array}$$
Applying the expression given by (Equation 18) of Definition 2 and relying on Lemma 1, the final result corresponds to the D-hypergeometric function of Lauricella $F_{D}^{\left(p\right)}\left(.\right)$ given by
(91)$$\begin{array}{c} E_{X^{1}}\left\{\ln\left[1+X^{T}\Sigma_{2}^{-1}X\right]\right\}=\psi \left(\frac{1+p}{2}\right)-\psi \left(\frac{1}{2}\right)\\ -\frac{\partial }{\partial a}\left\{\lambda_{p}^{-a}\sum_{\begin{array}{c} m_{1},\ldots ,\\ m_{p}=0 \end{array}}^{+\infty }\frac{{\left(a\right)}_{\sum_{i=1}^{p}m_{i}}{\left(a+\frac{1}{2}\right)}_{m_{p}}\prod_{i=1}^{p-1}{\left(\frac{1}{2}\right)}_{m_{i}}}{{\left(a+\frac{1+p}{2}\right)}_{\sum_{i=1}^{p}m_{i}}}\prod_{i=1}^{p-1}{\left(1-\frac{\lambda_{i}}{\lambda_{p}}\right)}^{m_{i}}\frac{1}{m_{i}!}\frac{{\left(1-\lambda_{p}^{-1}\right)}^{m_{p}}}{m_{p}!}\right\}{|}_{a=0} \end{array}$$
(92)$$\begin{array}{cc} \; & =\psi \left(\frac{1+p}{2}\right)-\psi \left(\frac{1}{2}\right)-\frac{\partial }{\partial a}\left\{\lambda_{p}^{-a}F_{D}^{\left(p\right)}\left(a,\underset{p}{\underbrace{\frac{1}{2},\ldots ,\frac{1}{2},a+\frac{1}{2}}};a+\frac{1+p}{2};1-\frac{\lambda_{1}}{\lambda_{p}},\ldots ,1-\frac{\lambda_{p-1}}{\lambda_{p}},1-\frac{1}{\lambda_{p}}\right)\right\}{|}_{a=0}. \end{array}$$
The final development of the previous expression is as follows
(93)$$\begin{array}{cc} \; & E_{X^{1}}\left\{\ln\left[1+X^{T}\Sigma_{2}^{-1}X\right]\right\}=\psi \left(\frac{1+p}{2}\right)-\psi \left(\frac{1}{2}\right)+\ln\lambda_{p}\\ \; & -\frac{\partial }{\partial a}\left\{F_{D}^{\left(p\right)}\left(a,\underset{p}{\underbrace{\frac{1}{2},\frac{1}{2},\ldots ,\frac{1}{2},a+\frac{1}{2}}};a+\frac{1+p}{2};1-\frac{\lambda_{1}}{\lambda_{p}},\ldots ,1-\frac{\lambda_{p-1}}{\lambda_{p}},1-\frac{1}{\lambda_{p}}\right)\right\}{|}_{a=0}. \end{array}$$ □

In this section, we presented the exact expression of $E_{X^{1}}\left\{\ln\left[1+X^{T}\Sigma_{2}^{-1}X\right]\right\}$. In addition, the multiple power series $F_{D}^{\left(p\right)}$ which appears to be a special case of $F_{N}^{\left(p\right)}$ provides many properties and numerous transformations (see Appendix A) that make easier the convergence of the multiple power series. In the next section, we establish the KLD closed-form expression based on the expression of the latter expectation.

## 6. KLD between Two Central MCDs

Plugging (Equation 39) and (Equation 93) into (Equation 5) yields the closed-form expression of the KLD between two central MCDs with pdfs $f_{X^{1}}\left(x|\Sigma_{1},p\right)$ and $f_{X^{2}}\left(x|\Sigma_{2},p\right)$. This result is presented in the following theorem.

**Theorem** **1.**
*Let $X^{1}$ and $X^{2}$ be two random vectors that follow central MCDs with pdfs given, respectively, by $f_{X^{1}}\left(x|\Sigma_{1},p\right)$ and $f_{X^{2}}\left(x|\Sigma_{2},p\right)$. The Kullback–Leibler divergence between central MCDs is*

(94)
$$\begin{array}{cc} \; & \mathrm{KL}(X^{1}\left|\right|X^{2})=-\frac{1}{2}\log\prod_{i=1}^{p}\lambda_{i}+\frac{1+p}{2}[\log\lambda_{p}\\ \; & -\frac{\partial }{\partial a}\left\{F_{D}^{\left(p\right)}\left(a,\underset{p}{\underbrace{\frac{1}{2},\ldots ,\frac{1}{2},a+\frac{1}{2}}};a+\frac{1+p}{2};1-\frac{\lambda_{1}}{\lambda_{p}},\ldots ,1-\frac{\lambda_{p-1}}{\lambda_{p}},1-\frac{1}{\lambda_{p}}\right)\right\}{|}_{a=0}] \end{array}$$

*where λ1,…, λp are the eigenvalues of the real matrix Σ1Σ2−1, and FD(p)(.) represents the Lauricella D-hypergeometric function defined for p variables.*


Lauricella [39] gave several transformation formulas (see Appendix A), whose relations (Equation 137)–(A7), and (A9) are applied to $F_{D}^{\left(p\right)}\left(.\right)$ in (Equation 94). The results of transformation are as follows
(95)$$\begin{array}{c} F_{D}^{\left(p\right)}\left(a,\frac{1}{2},\ldots ,\frac{1}{2},a+\frac{1}{2};a+\frac{1+p}{2};1-\frac{\lambda_{1}}{\lambda_{p}},\ldots ,1-\frac{\lambda_{p-1}}{\lambda_{p}},1-\frac{1}{\lambda_{p}}\right)\\ =\lambda_{p}^{a+\frac{p}{2}}\prod_{i=1}^{p-1}\lambda_{i}^{-\frac{1}{2}}F_{D}^{\left(p\right)}\left(\frac{1+p}{2},\frac{1}{2},\ldots ,\frac{1}{2},a+\frac{1}{2};a+\frac{1+p}{2};1-\frac{\lambda_{p}}{\lambda_{1}},\ldots ,1-\frac{\lambda_{p}}{\lambda_{p-1}},1-\lambda_{p}\right) \end{array}$$
(96)$$\begin{array}{cc} \; & ={\left(\frac{\lambda_{1}}{\lambda_{p}}\right)}^{-a}F_{D}^{\left(p\right)}\left(a,\frac{1}{2},\ldots ,\frac{1}{2},a+\frac{1}{2};a+\frac{1+p}{2};1-\frac{\lambda_{p}}{\lambda_{1}},\ldots ,1-\frac{\lambda_{2}}{\lambda_{1}},1-\frac{1}{\lambda_{1}}\right) \end{array}$$
(97)$$\begin{array}{cc} \; & =\lambda_{p}^{a}F_{D}^{\left(p\right)}\left(a,\frac{1}{2},\ldots ,\frac{1}{2};a+\frac{1+p}{2};1-\lambda_{1},1-\lambda_{2},\ldots ,1-\lambda_{p}\right) \end{array}$$
(98)$$\begin{array}{cc} \; & =\lambda_{p}^{a}\prod_{i=1}^{p}\lambda_{i}^{-\frac{1}{2}}F_{D}^{\left(p\right)}\left(\frac{1+p}{2},\frac{1}{2},\ldots ,\frac{1}{2};a+\frac{1+p}{2};1-\frac{1}{\lambda_{1}},1-\frac{1}{\lambda_{2}},\ldots ,1-\frac{1}{\lambda_{p}}\right). \end{array}$$
Considering the above equations, it is easy to provide different expressions of $\mathrm{KL}(X^{1}||X^{2})$ shown in Table 1. The derivative of the Lauricella D-hypergeometric series with respect to *a* goes through the derivation of the following expression
(99)$$\begin{array}{cc} \; & \frac{\partial }{\partial a}\left\{F_{D}^{\left(p\right)}\left(a,\frac{1}{2},\frac{1}{2},\ldots ,\frac{1}{2},a+\frac{1}{2};a+\frac{1+p}{2};1-\frac{\lambda_{1}}{\lambda_{p}},\ldots ,1-\frac{\lambda_{p-1}}{\lambda_{p}},1-\frac{1}{\lambda_{p}}\right)\right\}{|}_{a=0} \end{array}$$
(100)$$\begin{array}{cc} \; & =\sum_{\begin{array}{c} m_{1},\ldots ,\\ m_{p}=0 \end{array}}^{+\infty }\frac{\partial }{\partial a}\left\{\frac{{\left(a\right)}_{\sum_{i=1}^{p}m_{i}}{\left(a+\frac{1}{2}\right)}_{m_{p}}}{{\left(a+\frac{1+p}{2}\right)}_{\sum_{i=1}^{p}m_{i}}}\right\}{|}_{a=0}\prod_{i=1}^{p-1}{\left(\frac{1}{2}\right)}_{m_{i}}{\left(1-\frac{\lambda_{i}}{\lambda_{p}}\right)}^{m_{i}}\frac{1}{m_{i}!}\frac{{\left(1-\lambda_{p}^{-1}\right)}^{m_{p}}}{m_{p}!} \end{array}$$

The derivative with respect to *a* of the Lauricella D-hypergeometric series and its transformations goes through the following expressions (see Appendix B for demonstration)
(101)$$\begin{array}{cc} \; & \frac{\partial }{\partial a}\left\{\frac{{\left(a\right)}_{\sum_{i=1}^{p}m_{i}}{\left(a+\frac{1}{2}\right)}_{m_{p}}}{{\left(a+\frac{1+p}{2}\right)}_{\sum_{i=1}^{p}m_{i}}}\right\}{|}_{a=0}=\frac{{\left(\frac{1}{2}\right)}_{m_{p}}{\left(1\right)}_{\sum_{i=1}^{p}m_{i}}}{{\left(\frac{1+p}{2}\right)}_{\sum_{i=1}^{p}m_{i}}\left(\sum_{i=1}^{p}m_{i}\right)}, \end{array}$$
(102)$$\begin{array}{cc} \; & \frac{\partial }{\partial a}\left\{\frac{{\left(a\right)}_{\sum_{i=1}^{p}m_{i}}}{{\left(a+\frac{1+p}{2}\right)}_{\sum_{i=1}^{p}m_{i}}}\right\}{|}_{a=0}=\frac{{\left(1\right)}_{\sum_{i=1}^{p}m_{i}}}{{\left(\frac{1+p}{2}\right)}_{\sum_{i=1}^{p}m_{i}}\left(\sum_{i=1}^{p}m_{i}\right)}, \end{array}$$
(103)$$\begin{array}{cc} \; & \frac{\partial }{\partial a}\left\{\frac{{\left(a+\frac{1}{2}\right)}_{m_{p}}}{{\left(a+\frac{1+p}{2}\right)}_{\sum_{i=1}^{p}m_{i}}}\right\}{|}_{a=0}=\frac{{\left(\frac{1}{2}\right)}_{m_{p}}}{{\left(\frac{1+p}{2}\right)}_{\sum_{i=1}^{p}m_{i}}}\left[\sum_{k=0}^{m_{p}-1}\frac{1}{k+\frac{1}{2}}-\sum_{k=0}^{\sum_{i=1}^{p}m_{i}-1}\frac{1}{k+\frac{1+p}{2}}\right], \end{array}$$
(104)$$\begin{array}{cc} \; & \frac{\partial }{\partial a}\left\{\frac{1}{{\left(a+\frac{1+p}{2}\right)}_{\sum_{i=1}^{p}m_{i}}}\right\}{|}_{a=0}=\frac{-1}{{\left(\frac{1+p}{2}\right)}_{\sum_{i=1}^{p}m_{i}}}\sum_{k=0}^{\sum_{i=1}^{p}m_{i}-1}\frac{1}{k+\frac{1+p}{2}}. \end{array}$$

To derive the closed-form expression of $d_{\mathrm{KL}}\left(X^{1},X^{2}\right)$ we have to evaluate the expression of $\mathrm{KL}(X^{2}||X^{1})$. The latter can be easily deduced from $\mathrm{KL}(X^{1}||X^{2})$ as follows
(105)$$\begin{array}{cc} \; & \;\mathrm{KL}(X^{2}\left|\right|X^{1})=\frac{1}{2}\log\prod_{i=1}^{p}\lambda_{i}-\frac{1+p}{2}[\log\lambda_{p}\\ \; & +\frac{\partial }{\partial a}\left\{F_{D}^{\left(p\right)}\left(a,\frac{1}{2},\ldots ,\frac{1}{2},a+\frac{1}{2};a+\frac{1+p}{2};1-\frac{\lambda_{p}}{\lambda_{1}},\ldots ,1-\frac{\lambda_{p}}{\lambda_{p-1}},1-\lambda_{p}\right)\right\}{|}_{a=0}]. \end{array}$$
Proceeding in the same way by using Lauricella transformations, different expressions of $\mathrm{KL}(X^{2}||X^{1})$ are provided in Table 1. Finally, given the above results, it is straightforward to compute the symmetric KL similarity measure $d_{\mathrm{KL}}\left(X^{1},X^{2}\right)$ between $X^{1}$ and $X^{2}$. Technically, any combination of the $\mathrm{KL}(X^{1}||X^{2})$ and $\mathrm{KL}(X^{2}||X^{1})$ expressions is possible to compute $d_{\mathrm{KL}}\left(X^{1},X^{2}\right)$. However, we choose the same convergence region for the two divergences for the calculation of the distance. Some expressions of $d_{\mathrm{KL}}\left(X^{1},X^{2}\right)$ are given in Table 1.

**Table 1 entropy-24-00838-t001:** KLD and KL distance computed when $X^{1}$ and $X^{2}$ are two random vectors following central MCDs with pdfs $f_{X^{1}}\left(x|\Sigma_{1},p\right)$ and $f_{X^{2}}\left(x|\Sigma_{2},p\right)$.

KL(X1||X2)(106)=−12log∏i=1pλi+1+p2logλp−∂∂aFD(p)a,12,…,12,a+12⏟p;a+1+p2;1−λ1λp,…,1−λp−1λp,1−1λp|a=0(107)=−12log∏i=1pλi−1+p2λpp2∏i=1p−1λi−12∂∂aFD(p)1+p2,12,…,12,a+12⏟p;a+1+p2;1−λpλ1,…,1−λpλp−1,1−λp|a=0(108)=−12log∏i=1pλi+1+p2logλ1−∂∂aFD(p)a,12,…,12,a+12⏟p;a+1+p2;1−λpλ1,…,1−λ2λ1,1−1λ1|a=0(109)=−12log∏i=1pλi−1+p2∂∂aFD(p)a,12,…,12⏟p;a+1+p2;1−λ1,…,1−λp|a=0(110)=−12log∏i=1pλi−1+p2∏i=1pλi−12∂∂aFD(p)1+p2,12,…,12⏟p;a+1+p2;1−1λ1,…,1−1λp|a=0
KL(X2||X1)(111)=12log∏i=1pλi−1+p2logλp+∂∂aFD(p)a,12,…,12,a+12⏟p;a+1+p2;1−λpλ1,…,1−λpλp−1,1−λp|a=0(112)=12log∏i=1pλi−1+p2λp−p2∏i=1p−1λi12∂∂aFD(p)1+p2,12,…,12,a+12⏟p;a+1+p2;1−λ1λp,…,1−λp−1λp,1−1λp|a=0(113)=12log∏i=1pλi−1+p2logλ1+∂∂aFD(p)a,12,…,12,a+12⏟p;a+1+p2;1−λ1λp,…,1−λ1λ2,1−λ1|a=0(114)=12log∏i=1pλi−1+p2∂∂aFD(p)a,12,…,12⏟p;a+1+p2;1−1λ1,…,1−1λp|a=0(115)=12log∏i=1pλi−1+p2∏i=1pλi12∂∂aFD(p)1+p2,12,…,12⏟p;a+1+p2;1−λ1,…,1−λp|a=0
dKL(X1,X2)=1+p2[logλp−∂∂aFD(p)a,12,…,12,a+12⏟p;a+1+p2;1−λ1λp,…,1−λp−1λp,1−1λp|a=0−λp−p2∏i=1p−1λi12(116)×∂∂aFD(p)1+p2,12,…,12,a+12⏟p;a+1+p2;1−λ1λp,…,1−λp−1λp,1−1λp|a=0]=−1+p2[∂∂aFD(p)a,12,…,12︸p;a+1+p2;1−λ1,…,1−λp|a=0+∏i=1pλi12∂∂a{FD(p)(1+p2,12,…,12︸p;a+1+p2;(117)1−λ1,…,1−λp)}|a=0]=−1+p2[∏i=1pλi−12∂∂aFD(p)1+p2,12,…,12⏟p;a+1+p2;1−1λ1,…,1−1λp|a=0+∂∂a{FD(p)(a,12,…,12⏟p;a+1+p2;(118)1−1λ1,…,1−1λp)}|a=0]

## 7. Particular Cases: Univariate and Bivariate Cauchy Distribution

### 7.1. Case of p=1

This case corresponds to the univariate Cauchy distribution. The KLD is given by
(119)$$\begin{array}{cc} \mathrm{KL}(X^{1}||X^{2}) & =-\frac{1}{2}\log\lambda -\frac{\partial }{\partial a}\left\{{\;}_{2}F_{1}\left(a,\frac{1}{2};a+1;1-\lambda \right)\right\}{|}_{a=0} \end{array}$$
where ${}_{2}F_{1}$ is the Gauss’s hypergeometric function. The expression of the derivative of ${}_{2}F_{1}$ is given as follows (see Section C.1 for details of computation)
(120)$$\begin{array}{cc} \frac{\partial }{\partial a}\left\{{\;}_{2}F_{1}\left(a,\frac{1}{2};a+1;1-\lambda \right)\right\}{|}_{a=0} & =\sum_{n=1}^{\infty }{\left(\frac{1}{2}\right)}_{n}\frac{1}{n}\frac{{\left(1-\lambda \right)}^{n}}{n!}\\ \; & =-2\ln\left(\frac{1+\lambda^{1/2}}{2}\right). \end{array}$$

Accordingly, the KLD is then expressed as
(121)$$\begin{array}{cc} \mathrm{KL}(X^{1}||X^{2}) & =\log\frac{{\left(1+\lambda^{\frac{1}{2}}\right)}^{2}}{4\lambda^{\frac{1}{2}}} \end{array}$$
(122)$$\begin{array}{cc} \; & =\log\frac{{\left(1+\lambda^{-\frac{1}{2}}\right)}^{2}}{4\lambda^{-\frac{1}{2}}}=\mathrm{KL}(X^{2}\left|\right|X^{1}). \end{array}$$

We conclude that KLD between Cauchy densities is always symmetric. Interestingly, this is consistent with the result presented in [31].

### 7.2. Case of p=2

This case corresponds to the Bivariate Cauchy distribution. The KLD is then given by
(123)$$\begin{array}{cc} \; & \mathrm{KL}(X^{1}\left|\right|X^{2})=-\frac{1}{2}\log\lambda_{1}\lambda_{2}-\frac{3}{2}\frac{\partial }{\partial a}\left\{F_{1}\left(a,\frac{1}{2},\frac{1}{2};a+\frac{3}{2};1-\lambda_{1},1-\lambda_{2}\right)\right\}{|}_{a=0} \end{array}$$
where $F_{1}$ is the Appell’s hypergeometric function (see Appendix A). The expression of the derivative of $F_{1}$ can be further developed
(124)$$\begin{array}{cc} \; & \frac{\partial }{\partial a}\left\{F_{1}\left(a,\frac{1}{2},\frac{1}{2};a+\frac{3}{2};1-\lambda_{1},1-\lambda_{2}\right)\right\}{|}_{a=0}\\ \; & =\sum_{\begin{array}{c} n,m=0 \end{array}}^{+\infty }\frac{{\left(1\right)}_{m+n}{\left(\frac{1}{2}\right)}_{n}{\left(\frac{1}{2}\right)}_{m}}{{\left(\frac{3}{2}\right)}_{m+n}}\frac{1}{m+n}\frac{{\left(1-\lambda_{1}\right)}^{n}}{n!}\frac{{\left(1-\lambda_{2}\right)}^{m}}{m!}. \end{array}$$
In addition, when the eigenvalue $\lambda_{i}$ for i=1,2 takes some particular values, the expression of the KLD becomes very simple. In the following, we show some cases:

$\left(\lambda_{1}=1,\lambda_{2}\neq 1\right)$ or $\left(\lambda_{2}=1,\lambda_{1}\neq 1\right)$

For this particular case, we have
(125)$$\begin{array}{cc} \frac{\partial }{\partial a}\left\{F_{1}\left(a,\frac{1}{2},\frac{1}{2};a+\frac{3}{2};1-\lambda_{i},0\right)\right\}{|}_{a=0}\; & =\frac{\partial }{\partial a}\left\{{\;}_{2}F_{1}\left(a,\frac{1}{2};a+\frac{3}{2};1-\lambda_{i}\right)\right\}{|}_{a=0} \end{array}$$
(126)$$\begin{array}{cc} \; & =-\ln\lambda_{i}+\frac{1}{\sqrt{1-\lambda_{i}}}\ln\left(\frac{1-\sqrt{1-\lambda_{i}}}{1+\sqrt{1-\lambda_{i}}}\right)+2. \end{array}$$
The demonstration of the derivation is shown in Section C.2. Then, KLD becomes equal to
(127)$$\begin{array}{cc} \mathrm{KL}(X^{1}||X^{2}) & =\ln\lambda_{i}-\frac{3}{2}\frac{1}{\sqrt{1-\lambda_{i}}}\ln\left(\frac{1-\sqrt{1-\lambda_{i}}}{1+\sqrt{1-\lambda_{i}}}\right)-3. \end{array}$$



$\lambda_{1}=\lambda_{2}=\lambda$



For this particular case, we have
(128)$$\begin{array}{cc} \frac{\partial }{\partial a}\left\{F_{1}\left(a,\frac{1}{2},\frac{1}{2};a+\frac{3}{2};1-\lambda ,1-\lambda \right)\right\}{|}_{a=0}\; & =\frac{\partial }{\partial a}\left\{{\;}_{2}F_{1}\left(a,1;a+\frac{3}{2};1-\lambda \right)\right\}{|}_{a=0} \end{array}$$
(129)$$\begin{array}{cc} \; & =-\frac{2}{\sqrt{1-\lambda^{-1}}}\ln\left(\sqrt{\lambda }+\sqrt{\lambda -1}\right)+2. \end{array}$$
For more details about the demonstration see Section C.3. The KLD becomes equal to
(130)$$\begin{array}{cc} \mathrm{KL}(X^{1}||X^{2}) & =-\ln\lambda +\frac{3}{\sqrt{1-\lambda^{-1}}}\ln\left(\sqrt{\lambda }+\sqrt{\lambda -1}\right)-3. \end{array}$$
It is easy to deduce that
(131)$$\begin{array}{cc} \mathrm{KL}(X^{2}||X^{1}) & =\ln\lambda +\frac{3}{\sqrt{1-\lambda }}\ln\left(\sqrt{\lambda^{-1}}+\sqrt{\lambda^{-1}-1}\right)-3. \end{array}$$
This result can be demonstrated using the same process as $\mathrm{KL}(X^{1}||X^{2})$. It is worth to notice that $\mathrm{KL}(X^{1}\left|\right|X^{2})\neq \mathrm{KL}(X^{2}\left|\right|X^{1})$ which leads us to conclude that the property of symmetry observed for the univariate case is no longer valid in the multivariate case. Nielsen et al. in [32] gave the same conclusion.

## 8. Implementation and Comparison with Monte Carlo Technique

In this section, we show how we practically compute the numerical values of the KLD, especially when we have several equivalent expressions which differ in the region of convergence. To reach this goal, the eigenvalues of $\Sigma_{1}\Sigma_{2}^{-1}$ are rearranged in a descending order $\lambda_{p}>\lambda_{p-1}>\ldots >\lambda_{1}>0$. This operation is justified by Equation (Equation 53) where it can be seen that the permutation of the eigenvalues does not affect the expectation result. Three cases can be identified from the expressions of KLD.

### 8.1. Case $1>\lambda_{p}>\lambda_{p-1}>\ldots >\lambda_{1}>0$

The expression of $\mathrm{KL}(X^{1}||X^{2})$ is given by Equation (109) and $\mathrm{KL}(X^{2}||X^{1})$ is given by (115).

### 8.2. Case $\lambda_{p}>\lambda_{p-1}>\ldots >\lambda_{1}>1$

$\mathrm{KL}(X^{1}||X^{2})$ is given by the Equation (110) and $\mathrm{KL}(X^{2}||X^{1})$ is given by (114).

### 8.3. Case $\lambda_{p}>1$ and $\lambda_{1}<1$

This case guarantees that $0\leq 1-\lambda_{j}/\lambda_{p}<1$, j=1,…,p−1 and $0\leq 1-1/\lambda_{p}<1$. The expression of the $\mathrm{KL}(X^{1}||X^{2})$ is given by Equation (106) and $\mathrm{KL}(X^{2}||X^{1})$ is given by (112) or (113). To perform an evaluation of the quality of the numerical approximation of the derivative of the Lauricella series, we consider a case where an exact and simple expression of $\frac{\partial }{\partial a}\left\{F_{D}^{\left(p\right)}\left(.\right)\right\}{|}_{a=0}$ is possible. The following case where $\lambda_{1}=\ldots =\lambda_{p}=\lambda$ allows the Lauricella series to be equivalent to the Gauss hypergeometric function given as follows
(132)$$\begin{array}{cc} \; & F_{D}^{\left(p\right)}\left(a,\underset{p}{\underbrace{\frac{1}{2},\ldots ,\frac{1}{2}}};a+\frac{1+p}{2};1-\lambda ,\ldots ,1-\lambda \right)={\;}_{2}F_{1}\left(a,\frac{p}{2};a+\frac{1+p}{2};1-\lambda \right). \end{array}$$
This relation allows us to compare the computational accuracy of the approximation of the Lauricella series with respect to the Gauss function. In addition, to compute the numerical value the indices of the series will evolve from 0 to *N* instead of infinity. The latter is chosen to ensure a good approximation of the Lauricella series. Table 2 shows the computation of the derivative of $F_{D}^{\left(p\right)}\left(.\right)$ and ${}_{2}F_{1}\left(.\right)$, along with the absolute value of error |ϵ|, where p=2,N={20,30,40}. The exact expression of $\frac{\partial }{\partial a}{\{}_{2}F_{1}\left(.\right)\}{|}_{a=0}$ when p=2 is given by Equation (Equation 129). We can deduce the following conclusions. First, the error is reasonably low and decreases as the value of *N* increases. Second, the error increases for values of 1−λ close to 1 as expected, which corresponds to the convergence region limit.

In the following section, we compare the Monte Carlo sampling method to approximate the KLD value with the numerical value of the closed-form expression of the latter. The Monte Carlo method involves sampling a large number of samples and using them to calculate the sum rather than the integral. Here, for each sample size, the experiment is repeated 2000 times. The elements of $\Sigma_{1}$ and $\Sigma_{2}$ are given in Table 3. Figure 1 depicts the absolute value of bias, mean square error (MSE) and box plot of the difference between the symmetric KL approximated value and its theoretical one, given versus the sample sizes. As the sample size increases, the bias and the MSE decrease. Accordingly, the approximated value will be very close to the theoretical KLD when the number of samples is very large. The computation time of the proposed approximation and the classical Monte Carlo sampling method are recorded using Matlab on a 1.6 GHz processor with 16 GB of memory. For the proposed numerical approximation, the computation time is evaluated to 1.56 s with N=20. The value of *N* can be increased to further improve the accuracy, but it will increase the computation time. For the Monte Carlo sampling method, the mean time values at sample sizes of {65,536; 131,072; 262,144} are {2.71;5.46;10.78} seconds, respectively.

To further encourage the dissemination of these results, we provide a code available as attached file to this paper. This is given in Matlab with a specific case of p=3. This can be easily extended to any value of *p*, thanks to the general closed-form expression established in this paper.



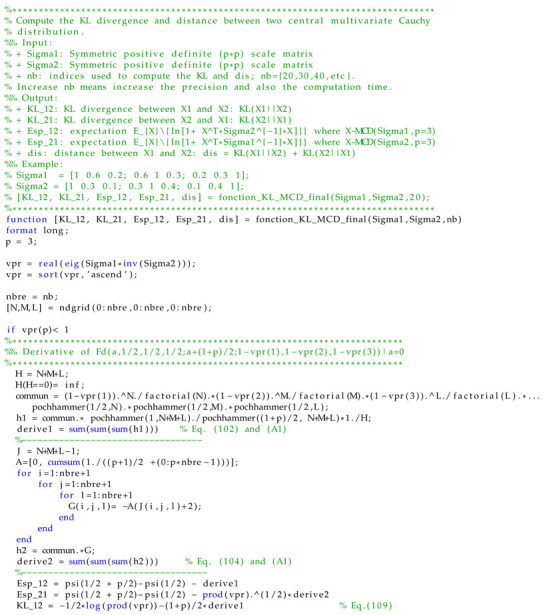





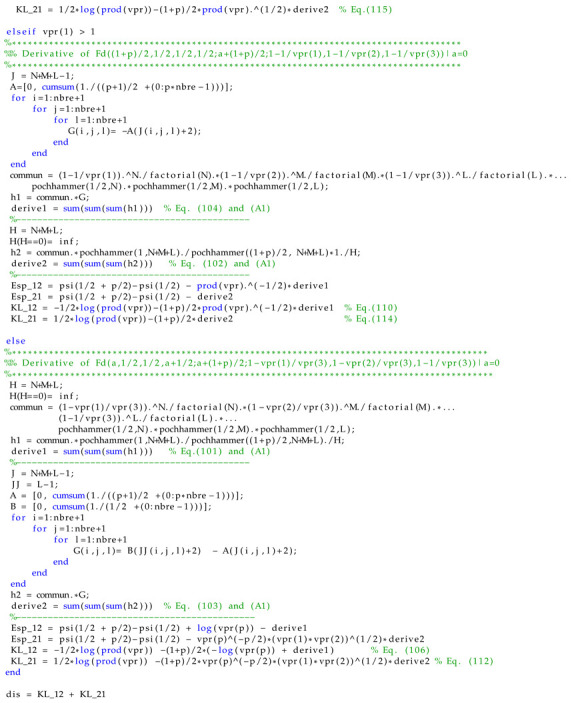



## 9. Conclusions

Since the MCDs have various applications in signal and image processing, the KLD between central MCDs tackles an important problem for future work on statistics, machine learning and other related fields in computer science. In this paper, we derived a closed-form expression of the KLD and distance between two central MCDs. The similarity measure can be expressed as function of the Lauricella D-hypergeometric series $F_{D}^{\left(p\right)}$. We have also proposed a simple scheme to compute easily the Lauricella series and to bypass the convergence constraints of this series. Codes and examples for numerical calculations are presented and explained in detail. Finally, a comparison is made to show how the Monte Carlo sampling method gives approximations close to the KLD theoretical value. As a final note, it is also possible to extend these results on the KLD to the case of the multivariate *t*-distribution since the MCD is a particular case of this multivariate distribution.

## Figures and Tables

**Figure 1 entropy-24-00838-f001:**
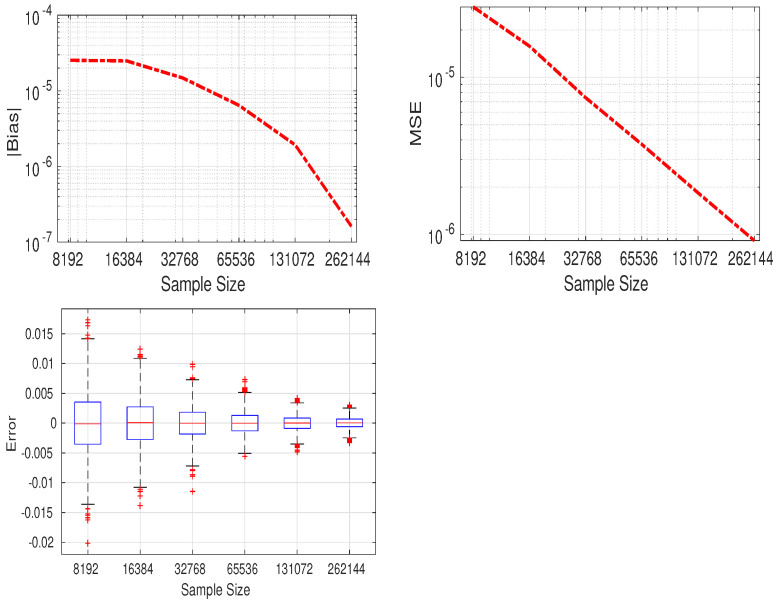
Top row: Bias (**left**) and MSE (**right**) of the difference between the approximated and theoretical symmetric KL for MCD. Bottom row: Box plot of the error. The mean error is the bias. Outliers are larger than $Q_{3}+1.5\times IQR$ or smaller than $Q_{1}-1.5\times IQR$, where $Q_{1}$, $Q_{3}$, and IQR are the 25th, 75th percentiles, and the interquartile range, respectively.

**Table 2 entropy-24-00838-t002:** Computation of $A=\frac{\partial }{\partial a}{\{}_{2}F_{1}\left(.\right)\}{|}_{a=0}$ and $B=\frac{\partial }{\partial a}\left\{F_{D}^{\left(p\right)}\left(.\right)\right\}{|}_{a=0}$ when p=2 and $\lambda_{1}=\ldots =\lambda_{p}=\lambda$.

		N=20		N=30		N=40	
1−λ	A	B	|ϵ|	B	|ϵ|	B	|ϵ|
0.1	0.0694	0.0694	9.1309$\;\times \;10^{-16}$	0.0694	9.1309$\;\times \;10^{-16}$	0.0694	9.1309$\;\times \;10^{-16}$
0.3	0.2291	0.2291	3.7747$\;\times \;10^{-14}$	0.2291	1.1102$\;\times \;10^{-16}$	0.2291	1.1102$\;\times \;10^{-16}$
0.5	0.4292	0.4292	2.6707$\;\times \;10^{-9}$	0.4292	1.2458$\;\times \;10^{-12}$	0.4292	6.6613$\;\times \;10^{-16}$
0.7	0.7022	0.7022	5.9260$\;\times \;10^{-6}$	0.7022	8.2678$\;\times \;10^{-8}$	0.7022	1.3911$\;\times \;10^{-9}$
0.9	1.1673	1.1634	0.0038	1.1665	7.2760$\;\times \;10^{-4}$	1.1671	1.6081$\;\times \;10^{-4}$
0.99	1.7043	1.5801	0.1241	1.6267	0.0776	1.6514	0.0529

**Table 3 entropy-24-00838-t003:** Parameters $\Sigma_{1}$ and $\Sigma_{2}$ used to compute KLD for central MCD.

Σ	$\Sigma_{11}$, $\Sigma_{22}$, $\Sigma_{33}$, $\Sigma_{12}$, $\Sigma_{13}$, $\Sigma_{23}$
$\Sigma_{1}$	1, 1, 1, 0.6, 0.2, 0.3
$\Sigma_{2}$	1, 1, 1, 0.3, 0.1, 0.4

## Data Availability

Not applicable.

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
