# Peer review of "A Generic Formula and Some Special Cases for the Kullback–Leibler Divergence between Central Multivariate Cauchy Distributions"

_entropy, 2022, doi:10.3390/e24060838_

Round 1
Reviewer 1 Report
In Paragraph 8 explain why authors decided to use N=20, 30. Please use greater N if possible.
In Conclusion suggest application of divergence between Cauchy distributions.
In Conclusion explains pros of results in comparison to paper 31 and 32 in references.
Author Response
Please find my responses to Reviewer 1 Comments in the attached file.

Reviewer 2 Report
Please check attached pdf

Reviewer 3 Report
The paper presents a calculation expression for the Kullback-Leibler divergence between two central multivariate Cauchy distributions. In order to obtain this expression, the authors use some specific tools based on multiple power series. Some particular cases, involving the univariate or the bivariate Cauchy distributions, are considered and discussed. Moreover, a procedure for numerical calculation of the Kullback-Leibler divergence expression is proposed.
The paper is interesting and represents a useful contribution, since the multivariate Cauchy distributions have applications in various problems concerning signal and image processing. The theoretical results are presented and explained in detail, codes and examples for numerical calculations illustrate the computational aspects. In my opinion this paper can be published in the Journal Entropy.
Author Response
Please find my responses to Reviewer 3 comments in the attached file
